



# Aerosol type classification analysis using EARLINET multiwavelength and depolarization lidar observations

Maria Mylonaki[1], Elina Giannakaki[2,3], Alexandros Papayannis[1], Christina-Anna Papanikolaou[1], Mika Komppula[3], Doina Nicolae[4], Nikolaos Papagiannopoulos[5,6], Aldo Amodeo[5], Holger Baars[7], Ourania Soupiona[1]

[1]Laser Remote Sensing Unit, Department of Physics, National and Technical University of Athens, Zografou, 15780, Greece
[2]Department of Environmental Physics and Meteorology, Faculty of Physics, National and Kapodistrian University of Athens, National and Kapodistrian University of Athens
[3]Finnish Meteorological Institute, P.O.Box 1627, 70211 Kuopio, Finland
[4] National Institute of R&D for Optoelectronics (INOE), Magurele, Romania
[5]Consiglio Nazionale delle Ricerche, Istituto di Metodologie per l'Analisi Ambientale (CNR-IMAA), C.da S. Loja, Tito Scalo (PZ), 85050, Italy
[6]CommSensLab, Dept. of Signal Theory and Communications, Universitat Politècnica de Catalunya, Barcelona, Spain
[7]Leibniz Institute for Tropospheric Research, Leipzig

*Correspondence to*: Maria Mylonaki (mylonaki.mari@gmail.com)

**Abstract.** We introduce an automated aerosol type classification method, called Source Classification ANalysis (SCAN). SCAN is based on predefined and characterized aerosol source regions, the time that the air parcel spends above each geographical region and a number of additional criteria. The output of SCAN is compared with two independent aerosol classification methods, which use the intensive optical parameters from lidar data: (1) "Mahalanobis distance automatic aerosol type classification" (MD) and (2) "Neural Network Aerosol Typing Algorithm" (NATALI). In this paper, data from the European Aerosol Research Lidar Network (EARLINET) have been used. A total of 97 free tropospheric (FT) aerosol layers from 4 typical EARLINET stations (i.e., Bucharest, Kuopio, Leipzig and Potenza) in the period 2014-2018 were classified based on a 3β+2α+1δ lidar configuration. We found that SCAN, being an optical property independent method, is not affected by the overlapping optical values of different aerosol types. Furthermore, SCAN has no limitations concerning its ability to classify different aerosol mixtures. Additionally, it is a valuable tool to classify aerosol layers, based on even to single (elastic) lidar signals, in case of lidar stations which cannot provide a full data set (3β+2α+1δ) of aerosol optical properties, therefore it can work independently of the capabilities of a lidar system. Finally, our results show that NATALI has the lower percentage of unclassified layers (4%), while MD has the percentage of unclassified layers (50%) and the lower percentage of cases classified as aerosol mixtures (5%).

## 1 Introduction

Aerosol particles affect directly the Earth's radiation budget by interacting, mainly, with the solar radiation through absorption and scattering (aerosol-radiation interaction, "ari") (Hobbs, 1993). Furthermore, aerosols affect cloud formation



and behaviour both serving as seeds (cloud condensation nuclei, ice nuclei) upon which cloud droplets and ice crystals form, and influencing the cloud albedo due to changing concentrations of cloud condensation and ice nuclei, also known as the

Twomey effect (aerosol-cloud interaction, "aci") (Twomey, 1959; Twomey and Warner, 1967; Hobbs et al., 1993; Stevens and Feigold, 2009; IPCC, 2014; Rosenfeld et al., 2014; Rosenfeld et al., 2016).

One accurate and powerful technique to study atmospheric aerosols is the light detection and ranging (lidar) which is based on the active remote sensing of the atmosphere (Weitcamp et al., 2005). This technique has received quite an attention, because of the multiple possibilities to retrieve near real-time information of the structure and the composition of the

atmosphere with high spatial (i.e. down to few meters) and temporal (i.e. down to seconds depending on the system) resolution. Specifically, the multi-wavelength Raman/depolarization lidars can be used for aerosol detection and characterization (i.e. dust, smoke, continental, etc.) as they provide vertically-resolved information of extensive [particle backscatter ($b_{\lambda\alpha}$) and extinction coefficients ($e_{\lambda\alpha}$)] and intensive aerosols properties [lidar ratio ($lr_{\lambda\alpha}$), Ångström exponent extinction- ($A_{e\lambda\alpha/\lambda\beta}$) and backscatter-related ($A_{b\lambda\alpha/\lambda\beta}$), particle linear depolarization ratio (pldr)] optical properties (Nicolae et

al., 2006; Burton et al., 2012; Groß et al., 2013; Giannakaki et al., 2016; Soupiona et al., 2019). Towards this direction, the large majority of the European Aerosol Research Lidar Network (EARLINET) stations is based on multi-wavelength Raman lidar systems, that combine detection channels at both elastic and Raman-shifted signals, and are equipped with depolarization channels (Pappalardo et al., 2014).

Until recently, the identification of aerosol layers was based, apart from the aerosol lidar data, on air mass back-trajectory

analysis, atmospheric models (e.g. DREAM; Basart et al., 2012), concurrent satellite products (MODIS dust and fire data; e.g., Giglio et al., 2013) as well as ground-based photometric data (Papayannis et al., 2005; Papayannis et al., 2008). It is well established that air-mass trajectory analysis (back to several hours) by using the HYSPLIT (Draxler et al., 1998), or FLEXPART (FLEXible PARTicle dispersion model; Stohl et al., 2005) models ending above the observation station is used in order to identify the air mass origin of the detected layer. However, this case-by-case aerosol layer identification is not

objective automated.

To overcome this defect, two automated methods have been developed recently to classify aerosol layers observed by lidars: 1) the "Mahalanobis distance aerosol classification algorithm" (MD) (Papagiannopoulos et al., 2018) which uses the lidar intensive properties ($lr_{\lambda\alpha}$, ratio of the lidar ratio ($lr_{\lambda 1}/lr_{\lambda 2}$), $A_{e\lambda\alpha/\lambda\beta}$, $A_{b\lambda\alpha/\lambda\beta}$ and lpdr (if provided)) in order to classify the measured aerosol layers into a number of aerosol types and 2) the "Neural Network Aerosol Classification Algorithm"

(NATALI) (Nicolae et al., 2018) which is based on artificial neural networks (ANNs) trained to estimate the most probable aerosol type from, solely, a set of multispectral lidar data (color index (ci), color ratio (cr), $lr_{\lambda\alpha}$, $A_{e\lambda\alpha/\lambda\beta}$, $A_{b\lambda\alpha/\lambda\beta}$ and lpdr (if provided)). Taking into account that both NATALI and MD experience several limitations as they request as input the aerosol intensive optical properties as stated previously (Nicolae et al., 2018; Voudouri et al., 2019), a more generic aerosol classification code free from these defects is needed.

Therefore, in this work, we develop an improved automated layer classification algorithm, based on air mass backward trajectory analysis and satellite data. The algorithm called "Source Classification Analysis" (SCAN) is based on the amount

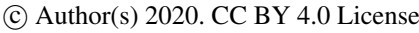


of time that the air parcel spends above certain already characterized aerosol source region and a number of additional criteria. This algorithm being aerosol optical property independent provides the advantage that its classification process is not affected by the overlapping values of the optical properties representing more than one aerosol types (eg. clean

continental, continental polluted, smoke). Furthermore, it has no limitations concerning its ability to classify aerosol mixtures. Finally, it can be useful for all types of lidar systems (independently of the number of channels used), as well as for other network-based systems (radar profilers, sunphotometers etc.).

In this paper, we use SCAN, MD and NATALI as classifiers to assign lidar observations to the pre-specified aerosol classes. The three different aerosol classification methods are described in Section 2, while Section 3 provides a discussion of the

results of the comparison between them. Finally, Section 4 provides the conclusions of our study.

## 2 Methodology

### 2.1 Aerosol Layer Classification algorithms

#### "Neural network aerosol classification algorithm"

NATALI (Nicolae et al., 2018) is an automated, optical property dependent, aerosol layer classification algorithm. The

typical aerosol profiles ($3b_{\lambda\alpha}+2e_{\lambda\alpha}+lpdr$ (optional), in netcdf format) from the EARLINET database are used as inputs in order to retrieve the mean aerosol optical properties within the layer boundaries indicated by the gradient method (Belegante et al., 2014). The learning process of the ANN has been performed using a synthetic database developed by Koepke et al., (1997) along with the T-matrix numerical method (Waterman, 1971; Mishchenko et al., 1996), to iteratively compute the intensive optical properties of 6 pure aerosol types (first 6 aerosol types in Table 1) and presented to the artificial neural

networks to perform the typing itself. The synthetic database is built for 350, 550, and 1000 nm wavelengths, which are then rescaled to the usual lidar wavelengths (i.e. 355, 532, and 1064 nm) using an Ångström exponent equal to 1. The mixtures are obtained by a linear combination of pure aerosol properties (Nicolae et al., 2018).

Two classification schemes are used with different aerosol type (classification) resolutions when particle depolarization data is available. The first one is applied when all the high-quality aerosol optical parameters are provided (uncertainty of $e_{\lambda\alpha} \leq 50$

%, uncertainty of $b_{\lambda\alpha} \leq 20$ %, uncertainty of lpdr $\leq 30$ %) and the aerosol typing is performed in the high resolution (AH) mode. This means that the aerosol mixtures can be sufficiently resolved providing the maximum number of outputs types (14 types, Table 1). In the second scheme, the values of the aerosol optical parameters have a high uncertainty (uncertainty of $e_{\lambda\alpha} > 50$ %, uncertainty of $b_{\lambda\alpha} > 20$ %, uncertainty of the lpdr $> 30$ %) and the typing is performed in the low resolution (AL) mode. In this case, the number of outputs types is limited to 6 (first 6 aerosol types, Table 1). A "voting" procedure selects

the most probable answer out of the two (possibly different) individual returns. The correct answer is selected based on a statistical approach considering two criteria: (i) which answer has a higher confidence and (ii) which answer is more stable over the uncertainty range (i.e. the percentage of agreement for values between error limits). Finally, there is the capability to



perform the typing when the particle depolarization is not available, knowing that the mixtures cannot be resolved, so only the predominant aerosol type is retrieved (Nicolae et al., 2018).

**"Mahalanobis distance aerosol classification algorithm"**

The automatic classification algorithm described in Papagiannopoulos et al. (2018) makes use of the Mahalanobis distance function that relates an unclassified measurement to a predefined aerosol type. The method compares the observations to model distributions that comprise EARLINET pre-classified data. Each Mahalanobis distance of an observation from a specific aerosol type is estimated, and the aerosol type is assigned for the minimum distance. Prior to the classification, two

screening criteria are assumed to ensure correct classification. The algorithm is able to classify an observation to a maximum of 8 (dust, volcanic, mixed dust, polluted dust, clean continental, mixed-marine, polluted continental, and smoke) and minimum of 4 (dust + volcanic + mixed dust + polluted dust, mixed-marine, smoke + polluted continental, and clean continental) aerosol classes depending on the lidar configuration.

In this study, we used four aerosol intensive properties: the backscatter-related Ångström exponent at 355 and 1064 nm, the

aerosol lidar ratio at 532 nm, the color ratio of the lidar ratios and the aerosol particle linear depolarization ratio at 532 nm, whie the minimum accepted distance was set to 4.3. As soon as the distance from a specific aerosol class is below the threshold and the remaining distances are higher than the threshold, the observation is assigned to that aerosol class. If more than one distance is below the threshold, the normalized probability of each class needs to be over 50%.

This objective multi-dimensional classification scheme has found great applicability and has been used with lidar (e.g.,

Burton et al., 2012; Papagiannopoulos et al., 2018), space-based polarimetry (Russell et al., 2014), and spectral photometry (e.g., Hamill et al., 2020; Siomos et al., 2020) data.  Kavoudou et al. (2019) compared NATALI and the Mahalanobis distance classification algorithm for the EARLINET station of Thessaloniki. Their study used a 3+2 lidar configuration and 4 aerosol classes (i.e., dust, maritime, polluted smoke, and clean continental) for each automatic algorithm. In general, they found fair agreement between MD and NATALI and the differences were attributed to the class definition and the range of

the class intensive properties.

**"Source classification analysis"**

SCAN is the automated aerosol layer classification process, optical property independent and developed in the IDL programming language in the frame of this study. For each identified aerosol layer a X-hours HYSPLIT backward trajectory (Draxler et al., 1998) is used to calculate the amount of time travelled above predefined aerosol source regions arriving over

a lidar station at the specific date and height that the aerosol layer is observed. X is the number of hours of the backward trajectory which can be decided by the user at the beginning of the process. SCAN assumes specific regions (Fig. 1, coloured squares, from now on mentioned as domains) in terms of aerosol sources (Penning de Vries et al., 2015).

Taking into account the information (latitude, longitude and height) from each HYSPLIT air-mass backward trajectory, SCAN implements a number of criteria: (i) if the geographical coordinates for the specific hour of the backward trajectory





are within the boundaries of the marine domains and if the height of this trajectory over the this domain is below 1 km (Wu et a;., 2008; Ho et al., 2015), SCAN assigns this layer to the "marine" aerosol type. (ii) If the geographical coordinates for the specific hour of the backward trajectory are within the boundaries of the clean continental, polluted continental or dust domains and if the height of this backward trajectory is below 2 km over the domain, SCAN assigns the specific hour to the "clean continental", "polluted continental" or "dust" aerosol type, respectively. (iii) For an hour to be assigned to the

"smoke" aerosol type, except from the coordinates of the backward trajectory at this specific hour, which should be within the boundaries of the "clean continental" or "polluted continental" domains, the height of the trajectory at this specific hour should be below 3 km (Amiridis et al., 2010).

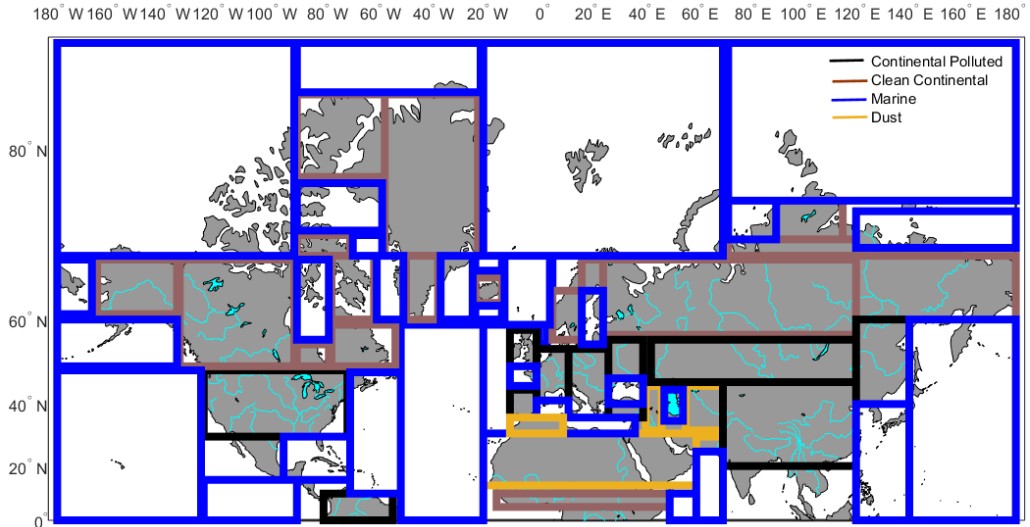

**Figure 1: SCAN's aerosol source type classification map. Different color squares represent different aerosol sources: orange**
**squares correspond to dust, blue squares to marine, brown squares to clean continental and black squares to continental polluted**
**aerosol sources.**

SCAN draws fire information from the Fire Information for Resource Management System (FIRMS) (https://firms.modaps.eosdis.nasa.gov) using the information of actively burning fires, along with their location, time and confidence value (in %) (Kaufman et al., 1998; Giglio et al., 2002) derived from Moderate Resolution Imaging
Spectroradiometer (MODIS) data. The selected time period is in accordance with HYSPLIT simulations (air mass backward duration). From SCAN's point of view, a hotspot is assumed as significant if the MODIS 'confidence' value is higher than 80% (Amiridis et al., 2010). In addition to the above criteria, the location of HYSPLIT air mass backward trajectory at the specific hour must have a maximum distance of 8 km away from a hotspot of high confidence, in order to be assigned as "smoke" aerosol type.
SCAN performs the above classification process for all the HYSPLIT air-mass backward trajectories, and as a final step, it counts the hours that the air parcel spends above each geographical domain. If more than one domain is involved following





the backward trajectory's path, a mixture of more than one aerosol type is assumed. In case the aforementioned criteria (domain and height limitations) are not satisfied, the aerosol type is considered 'unknown'.

The maximum number of pure aerosol types that SCAN can assign to a layer is 6 and the combination of them gives to
SCAN the capability to indentify aerosols from different sources in a specific layer. In Table 1 one can find the pure aerosol types and the mixtures that SCAN has dealt with in this study. The whole classification procedure of SCAN is displayed in the flowchart of Fig. 2.

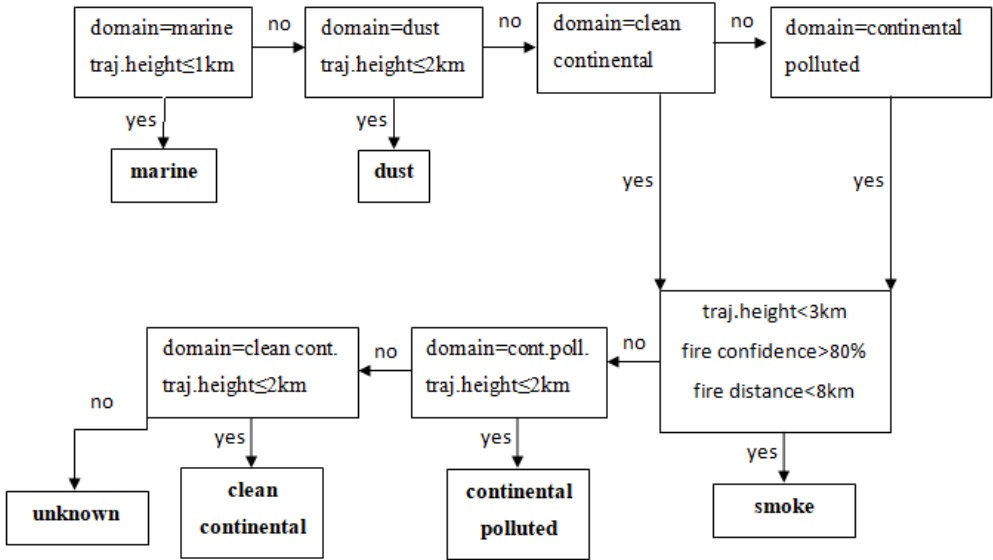

**Figure 2: Classification procedure of SCAN. This procedure is performed for each hour of the HYSPLIT back trajectory**
**associated with each layer.**

## 2.2 EARLINET lidar stations and data

The lidar station selection was based on the availability of the vertical profiles of a full set ($3b_{\lambda\alpha}+2e_{\lambda\alpha}+lpdr$) of aerosol optical properties at several wavelengths: backscatter coefficient ($b_{355}$, $b_{532}$, $b_{1064}$), extinction coefficient ($e_{355}$, $e_{532}$), lidar ratio ($lr_{355}$, $lr_{532}$), Ångström exponent ($A_{e355/532}$, $A_{b355/532}$, $A_{b532/1064}$) and particle linear depolarization ratio ($pldr_{532}$) at the EARLINET
database during the period 2014-2018. Therefore, the selected lidar stations are: Bucharest, Romania; Kuopio, Finland; Leipzig, Germany; and Potenza, Italy (Table 2). Thus, 48 full sets ($3b_{\lambda\alpha}+2e_{\lambda\alpha}+pldr$) of aerosol optical properties of lidar observations from the aforementioned stations have been studied. For each data set, and each aerosol layer, the geometrical layer boundaries (bottom, top) have been calculated according to Belegante et al. (2014). In total, 97 FT aerosol layers were obtained and their mean aerosol optical properties (intensive and extensive) were calculated.





## 2.3 Case studies

In this section 4 selected atmospheric layers involving different types of probed aerosols are presented. The performance of the 3 automated aerosol typing algorithms to aerosol classification is discussed in detail.

**Figure 3: Vertical profiles of aerosol optical properties (a) b355, b532, b1064, (b) e355, e532, (c) lr355, lr532, (d) Ae355/532, Ab355/532, Ab532/1064, (e) vdr532 and pldr532 observed over (i) Kuopio (19:25 UTC) and (ii) Potenza (21:26 UTC) on the 30th of July 2015, along with their mean values and standard deviations (inserted text).**

Figure 3 illustrates the vertical profiles of the aerosol optical properties, along with the mean values and standard deviations (inserted text) of (3a) $b_{355}$, $b_{532}$, $b_{1064}$ [$sr^{-1}$ $Mm^{-1}$], (3b) $e_{355}$, $e_{532}$ [$Mm^{-1}$], (3c) $lr_{355}$, $lr_{532}$ [$sr$], (3d) $A_{e355/532}$, $A_{b355/532}$, $A_{b532/1064}$


and (3e) the $pldr_{532}$ [%] of the aerosol layer observed on the 30th of July 2015 over (i) Kuopio (19:25 UTC), and (ii) over Potenza (21:26 UTC). The bottom and top boundaries of the aerosol layer observed over Kuopio are estimated at 1.5 and 1.9 km amsl. (Fig. 3i, lower and upper horizontal red line), respectively. For the same day, 3 different aerosol layers are detected over Potenza with corresponding values of bottom and top of (Fig. 3ii), (1) 2.8 and 3.1 km amsl. (lower layer), (2) 3.4 and 3.9 km (middle layer) and (3) 4.5 and 5.4 km (upper layer), respectively.

The mean values of the intensive aerosol optical properties within the aerosol layer observed over Kuopio are: $lr_{355}$=65.58±11.02 sr, $lr_{532}$=72.51±17.61 sr, $A_{e355/532}$=1.23±0.62, $A_{b355/532}$=1.36±0.0.05, $A_{b532/1064}$=1.23±0.05 and $pldr_{532}$=2.1±0.1 %, indicating fine absorbing aerosols. The mean values of the intensive aerosol optical property within the lower aerosol layer observed over Potenza on the same day are: $lr_{355}$=35.97±1.09 sr, $lr_{532}$=24.55±4.15 sr, $A_{e355/532}$=1.14±0.44, $A_{b355/532}$=0.16±0.04, $A_{b532/1064}$=0.97±0.04, and $pldr_{532}$=13.5±0.4 %. Similarly, for the middle aerosol layer observed over

Potenza: $lr_{355}$=31.05±2.11 sr, $lr_{532}$=22.50±1.69 sr, $A_{e355/532}$=0.86±0.25, $A_{b355/532}$=0.06±0.13, $A_{b532/1064}$=0.79±0.06 and $pldr_{532}$=15.4±1.5 %. These values of both lower and middle aerosol layers are indicative of coarse semi-depolarizing aerosols, probably, mixed dust particles. Finally, for the upper aerosol layer observed over Potenza: $lr_{355}$=38.77±4.81 sr, $lr_{532}$=24.44±3.39 sr, $A_{e355/532}$=0.56±0.37, $A_{b355/532}$=-0.58±0.25, $A_{b532/1064}$=0.72±0.07 and $pldr_{532}$=24.8±1.0 %, indicating coarse high-depolarizing aerosols, probably of dust origin.

Figure 4 illustrates the 6 days (144 hours) backward trajectory analysis for air masses ending on the 30th of July 2015 over the station of (i) Kuopio (62.74° N, 27.54° E) at 1500 m amsl. (19:00 UTC) and (ii) over Potenza (40.60° N, 15.72° E) at (1) 3000 m amsl., (2) 3800 m amsl. and (3) 5000 m amsl. (21:00 UTC), respectively. The colour bar indicates the trajectory's height amsl. for each hour of its journey. According to Fig. 4i the air masses that reached Kuopio on that day, at 1500 m amsl. (19:00 UTC), travelled from Ireland to Southern Finland, from 26/07 to 30/07, at ~1500 m amsl. and were probably

affected by continental polluted and clean continental aerosol sources of Northern Europe. The same air masses seem to be affected, also, by marine aerosols from 24/07 to 25/07, when travelling at lower heights (<1000 m amsl.) over Southern Ireland.

On the contrary, the air masses that reached Potenza on the 30th of July 2015 at 3000 m amsl. (lower layer, Fig. 4ii) travelled at a height of ~2000 m amsl. throughout its 6-days journey. These air masses started from North-Western Africa remained in

the area almost 3 days, and then passed over Southern Spain before reaching Potenza. The aerosol layer observed over Potenza the same date and hour at 3800 m amsl. (middle layer, Fig. 3ii, (a-b)) originated over the Northern Atlantic Ocean at a height of ~5000 m amsl. six days before and slowly descended at lower altitudes before passing over Northern Spain near ground level (Fig. 4ii, (2)). In the following two days, the air mass travelled at a height of 2000-4000 m amsl. from Spain to Italy over the Mediterranean Sea and before reaching Potenza.

Finally, the upper aerosol layer observed over Potenza at 5000 m amsl. (upper layer, Fig. 3ii, (a-b)) had a similar origin with the previous one expect the three first days of its journey when the air masses travelled over Atlantic Ocean at low altitudes (<1000 m amsl.) enhancing the  marine contribution to that layer (Fig. 4ii, (3)).

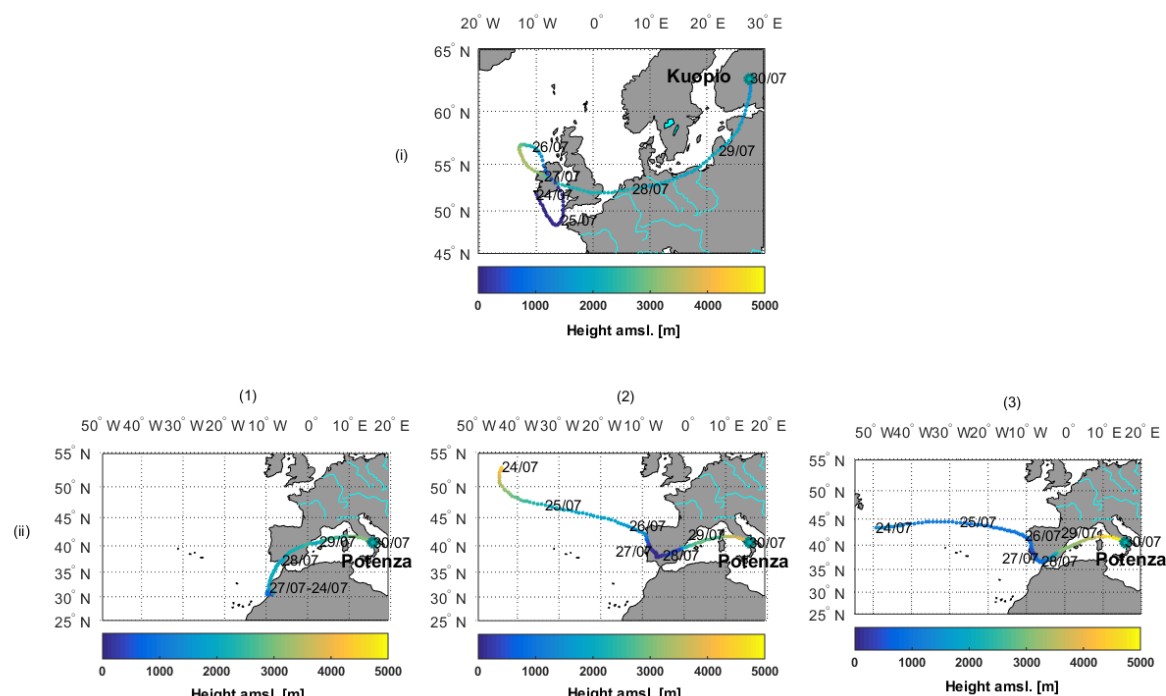

**Figure 4: HYSPLIT (GDAS Meteorological Data) 6 days (144 hours) air mass backward trajectory ending on the 30th of July 2015 over the lidar station of (i) Kuopio (62.74° N, 27.54° E) at 1500 m. amsl. (19:00 UTC), and (ii) Potenza (40.60° N, 15.72° E) at (1) 3000 m. amsl., (2) 4000 m. amsl., (3) 5000 m. amsl. (21:00 UTC), using Model Vertical Velocity as Vertical Motion Calculation Method. The colour bar indicates the trajectory's height above mean sea level for each hour of its journey.**

In Fig. 5 we present the classification of the aerosol layers under study, by (a) NATALI, (b) MD and (c) SCAN. The "aerosol type" results are given, concerning the classification by NATALI. Concerning the MD classification, the possibilities of each aerosol type assumed by MD are, also, given. Finally, concerning the classification of the aerosols by SCAN, the time (in hours) within which the air mass circulated over specified domains is also provided. It should be noted here that different colours refer to different aerosol types or aerosol mixtures.

Thus, in Fig. 5 we observe that NATALI classified the aerosol layer observed over Kuopio as "Continental Polluted". MD, also, gave the highest probability (65%) for the same layer to be of "Continental Polluted" origin, and typed it as such, while the second closest probability was 11% "Smoke". Finally, SCAN attributed 24 hours to that air mass and classified it as "Continental Polluted". Moreover, 17 hours were attributed as "Clean Continental" and 2 hours as "Marine" out of the 144 hours of the air mass backward trajectory, so, finally, was classified as cp+cc+m mixture (Fig. 5i, (a)-(b)-(c)). For the 101 remaining hours of the air mass backward trajectory that SCAN did not take into account, it was assumed that the air mass travelled without being affected by any aerosol source, as a result of the combination of the height and domain limitations, criteria that SCAN takes into account during its classification process. Taking into account these criteria and the air mass back-trajectory provided by HYSPLIT, we would expect that SCAN should have counted more hours to that air mass, attributed as "Marine". This is because of the predefined domains on the map as possible aerosol sources, which reduces the





spatial accuracy of the classification method (Fig. 1). Therefore, we can see that the final results of the 3 methods (Fig. 5i, (a)-(b)-(c)) are in good agreement concerning the layer observed over Kuopio, although, MD and SCAN can provide additional information of the constituents of the aerosol layer.

**Figure 5: Aerosol layers observed over (i) Kuopio and (ii) Potenza at 3000 m amsl. (bottom), 3800 m amsl. (middle), 5000 m amsl. (top), on the 30th of July 2015 classified by (a) NATALI, (b) MD and (c) SCAN.**

On the other hand, the classification results for the aerosol layers observed over Potenza on the 30th of July 2015 are more complex (Fig. 3, (ii)). NATALI classified the lower and middle aerosol layers as "Marine/CC" and the upper one as "Mineral Mixtures/Volcanic". However, regarding the lower and middle aerosol layers, it is highly unlike to be of "Marine"





origin, as they are not affected by the sea spray at these heights (> 2.5 km amsl.). These erroneous classification results must
have been affected by the low lr values (22-35±5 sr).

In this case MD gave to the aforementioned lower aerosol layer the probability of 41% to be of "Continental Polluted" origin, while there was also a 22% probability for the layer to be "Clean Continental". MD also, classified the middle aerosol layer as "Marine" with a 63% probability, while it seems to attribute a 16% probability to the "Continental Polluted" aerosol type. Again, the classification results for the lower and the middle layers have probably been affected by the low lr values.
Concerning the upper aerosol layer observed over Potenza, MD failed to characterize it, as it gave nearly equal probabilities of its possible aerosol types.

Taking, now, into account the results provided by SCAN concerning the lower aerosol layer observed above Potenza, SCAN counted 78 hours as "Dust" and 11 hours as "Continental Polluted" and finally classifying the aerosols as a mixture of "Continental Polluted+Dust". Concerning the middle aerosol layer, SCAN counted 64 hours as "Continental Polluted"
aerosols (out of 144 total hours), while one would expect a contribution of dust aerosols, as well, according to both $lpdr_{532}$ values and the origin of the air masses based on the air mass backward trajectory analysis. This indicates, again, that the predefined domains on the map of possible aerosol sources reduce the spatial accuracy of the classification method, especially when it comes to the Southern Mediterranean Sea in the vicinity of the Saharan desert. In these cases, an atmospheric dust model (e.g. BSC Dream model) should be used synergistically. Finally, concerning the upper aerosol layer
observed above Potenza, SCAN counted 66 hours as "Continental Polluted" aerosols and 44 hours as "Marine" aerosols, the latter being again highly impossible to happen at that height (~5 km amsl.), as previously explained. Again, here, it becomes obvious that an atmospheric dust model should be used synergistically with the SCAN results when we deal with air mass backward trajectories passing over the Southern Mediterranean Sea, due to the vicinity of the Saharan desert.

## 3 Results

So far, 97 FT aerosol layers have been classified by the aforementioned three classification algorithms. The results have been separated in aerosol layers according to aerosol types, as follows: The "1 type" (Fig. 6, blue) and "mixture" (Fig. 6, cyan) categories represent the aerosol layers that consist of one and two or more aerosol types, respectively. The "other" category consists of cases that NATALI marked as 'aerosol type/cloud contaminated' (i.e. marine/cc) and finally, "unknown" category (Fig. 6, yellow) consists of the cases that the method was unable to identify the source of the observed aerosol
layers. All the aerosol types and mixtures considered inside each algorithm is demonstrated in Table 1.





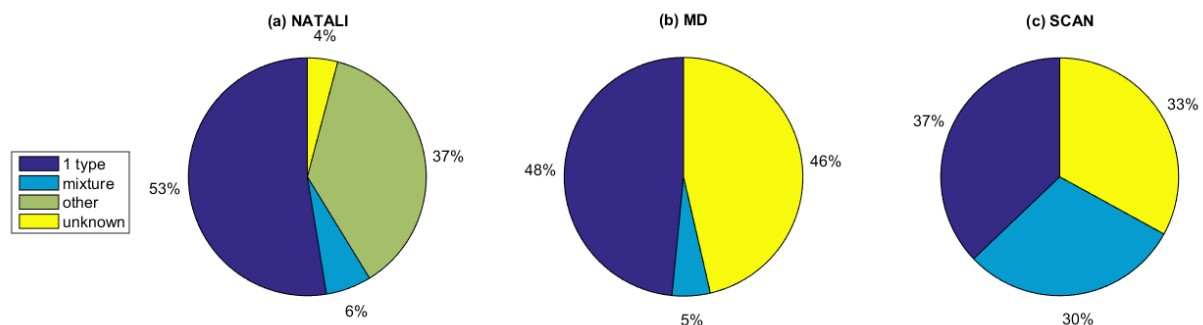

**Figure 6: Percentages of classified aerosol layers by (a) NATALI, (b) MD, (c) SCAN. Blue: 1 type aerosol layers, cyan: mixtures, green: other types and yellow: unknown constitution of aerosol layers.**

It can be concluded that NATALI (Fig. 6a) is able to classify the highest number of cases (94 cases), while MD (Fig. 6b)

failed to classify the highest number of cases (46%) and the lower percentage in classifying aerosols as of "mixture" types, which is a reasonable outcome considering that MD scheme considers only 2 aerosol mixtures, while NATALI and SCAN have way more. Finally, the SCAN algorithm (Fig. 6c) classified the 37% of the aerosol layers (36 layers) as "1 type", 30% (29 layers) as aerosol "mixtures", and 32% (32 layers) as "unknown" types.

## 3.1 Comparison of aerosol classification codes

In Fig. 7 we present the comparison of the classification results obtained using the pairs of (a) NATALI and MD, (b) MD and SCAN and (c) SCAN and NATALI. The number of aerosol layers classified as indicated by the row 'i' and the column 'j' is given inside each '(i,j)' square. For example, the number of aerosol layer classified as being of "1 type" by MD, but as "mixture" of aerosols by SCAN, is shown inside the ("1 type"-"mixture") square (16 such cases for this example in Fig. 7b).





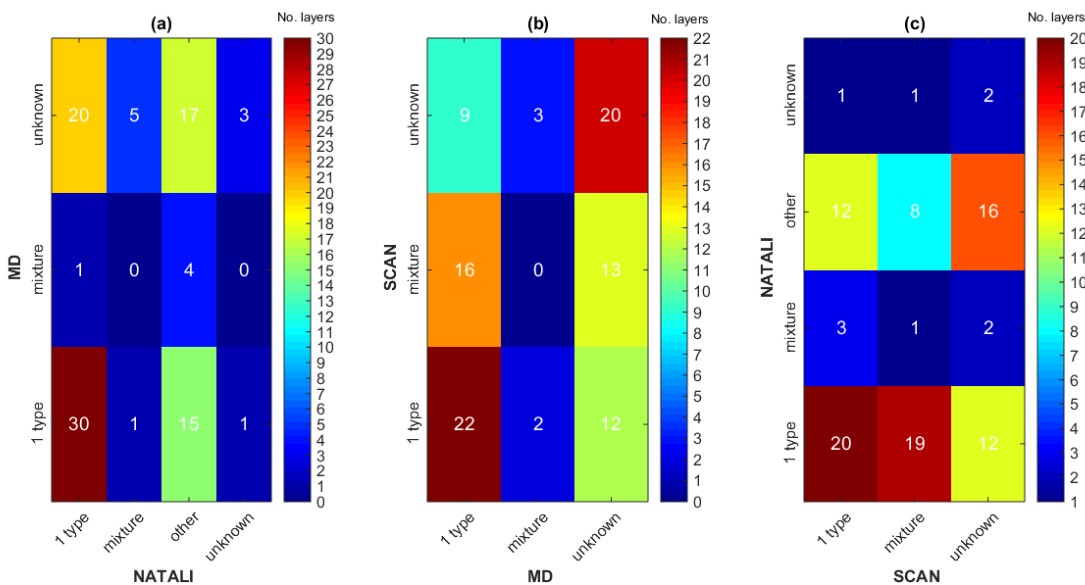


**Figure 7: Comparison between classified aerosol layers (a) NATALI and MD, (b) MD and SCAN and (c) SCAN and NATALI. The number of classified aerosol layers as indicated by the 'i' row and the 'j' column is given inside each '(i,j)' square.**

In Fig. 7a it can be seen that the 45 cases classified as "unknown" cases by MD, were either of "1 type" (20 cases) or "mixture" (5 cases) and "other" (17 cases) by NATALI and of "1 type" (12 cases) or "mixture" (13 cases) by SCAN (Fig.

7b) Furthermore, it seems that MD is unable to discriminate the different aerosol types inside the "other" layers according to NATALI (Fig. 7a) and the "mixture" layers according to SCAN, labelling them as "1 type" (Fig. 7b), which is probably because only two aerosol mixtures are considered inside MD. Concerning the 29 "unknown" cases classified by SCAN, 20 of them were also identified as "unknown" by MD (Fig. 7b), and were almost equally separated to "1 type" (12 cases) and "other" (16 cases) by NATALI (Fig. 7c).

**3.1.1 NATALI versus MD**

Figure 8a presents the comparison between the cases that MD classified as "1 type" or "mixture" against those classified as "other" by NATALI, while Fig. 8b shows the number of aerosol layers that MD classified as "unknown",and as "1 type" by NATALI. Finally, in Fig. 8c we present the number of aerosol layers that MD classified as "unknown" and as "mixture" or "other" by NATALI. The pie charts above each bar (Fig. 8b) and each stem (Fig. 8c) reveal the mean frequencies of each

aerosol type as calculated by MD.





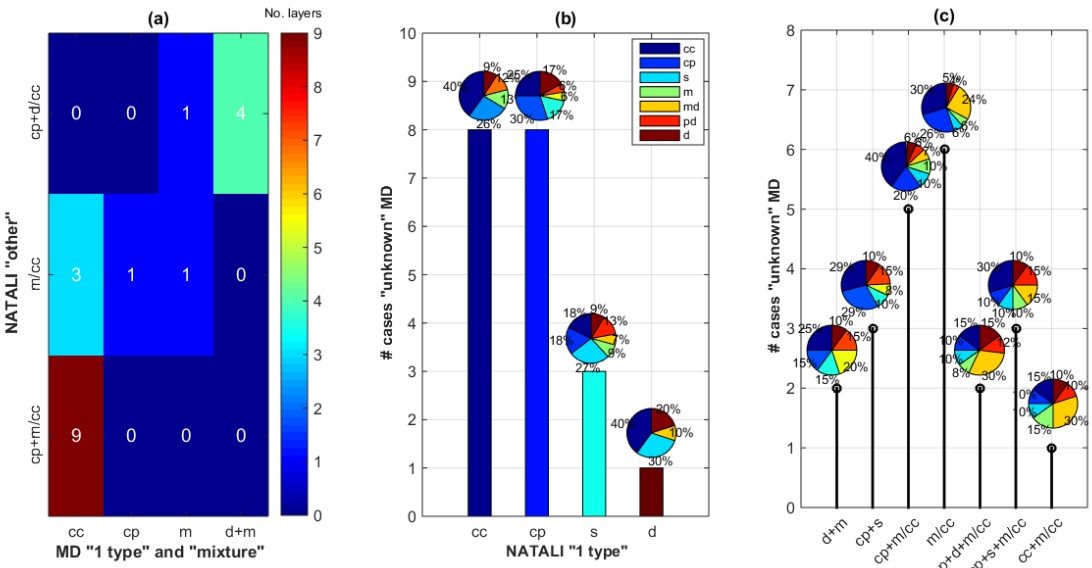

**Figure 8: (a)** Comparison between the cases classified by MD as "1 type" or "mixture" and as "other" by NATALI. **(b)** Number of aerosol layers classified by MD as "unknown" and as "1 type" by NATALI. **(c)** Number of aerosol layers classified by MD as "unknown" and as "mixture" or "other" by NATALI. The pie charts above each bar and each stem reveal the mean frequencies of each aerosol type as calculated by MD.

The inability of MD to classify the aerosol layers according to NATALIS's classification (Fig. 8a) can be attributed to the characteristics of the aerosol layers not well-modelled by the algorithm, which means that the intensive parameters are not within the accepted "boarders" of the pre-defined classes of the algorithm. Concerning the "1 type" aerosol layers according to NATALI (Fig. 8b), MD would have predicted them correctly if the labeling of these layers by MD was achieved considering the higher percentage of the aerosol types, as the pie charts above each bar indicate (Fig 8b). This does not seem to be the case for the "Dust" type labeled by NATALI to which MD gave 40% probability to be "Continental Polluted", 30% "Smoke" and only 30% "Dust" and "Mixed Dust (Fig. 8b). Concerning the "mixture" and "other" according to NATALI (Fig. 8c), it seems that MD found a high contribution of dust and mixtures of dust aerosols (approximately 50%) inside these layers (yellow, orange and dark red aerosol types according to pie charts above the stems).

### 3.1.2 MD versus SCAN

Figure 9a presents the comparison between the cases that were classified by MD as "1 type" and as "mixture" by SCAN, while Fig. 9b shows the number of aerosol layers that were classified by MD as "unknown" and as "1 type" by SCAN. Finally, Fig. 9c presents the number of aerosol layers classified by MD as "unknown", and as "mixture" or "other" by SCAN. The pie charts above each bar (Fig. 9b) and each stem (Fig. 9c) reveal the mean frequencies of each aerosol type as calculated by MD.



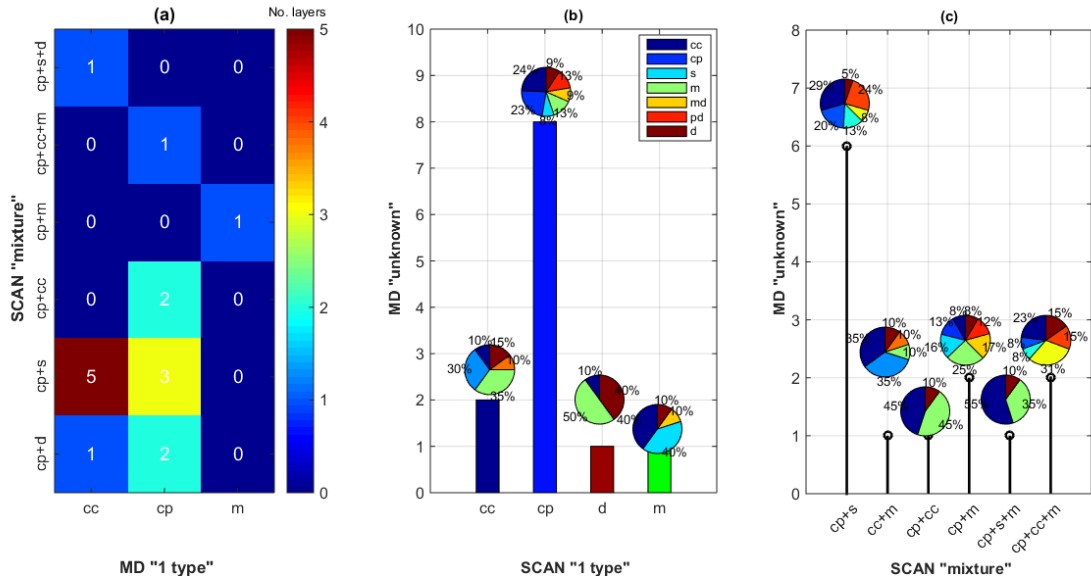

**Figure 9: (a) Comparison between the cases that were classified by MD as "1 type" and as "mixture" by SCAN. (b) Number of aerosol layers that were classified by MD as "unknown" and as "1 type" by SCAN. (c) Number of aerosol layers that were classified by MD as "unknown" and as "mixture" or "other" by SCAN. The pie charts above each bar and each stem reveal the mean frequencies of each aerosol type as calculated by MD.**

The misclassification of an aerosol layer by MD compared to the classification as "Continental Polluted and Smoke" layer by SCAN (Fig. 9a) could be, again, attributed to the location of the observation compared to the location of the predefined aerosol types of MD classification algorithm which depends on the aerosol optical properties of the studied layers. Additionally, it seems that the aerosol optical properties of the mixture "Continental Polluted and Smoke" by SCAN are attributed either to "Clean Continental" or to "Continental Polluted" aerosols, by MD (Fig. 9a). From the cases that MD classified as "unknown", 8 are classified as "Continental Polluted" (cp) by SCAN (Fig. 9b), while other 6 cases are classified as "Continental Polluted and Smoke" (cp+s) (Fig. 9c). Finally, from these "unknown" cases by MD, there are other 11 cases that SCAN classified as "Clean Continental" (2 cases), "Dust" (one case), "Marine" (1 case), (Fig. 9b), or "Clean Continental and Marine" (1 case), "Continental Polluted and Clean Continental" (1 case), "Continental Polluted and Marine" (2 cases), "Continental Polluted, Smoke and Marine" (1 case) and "Continental Polluted, Clean Continental and Marine" (2 cases) (Fig. 9c).

### 3.1.3 SCAN versus NATALI

Figure 10 presents the comparison between the cases that (a) SCAN classified as "mixture" and NATALI as "1 type", (b) SCAN classified as "1 type" and NATALI as "other" and (c) SCAN classified as "mixture" and NATALI as "other" by NATALI.



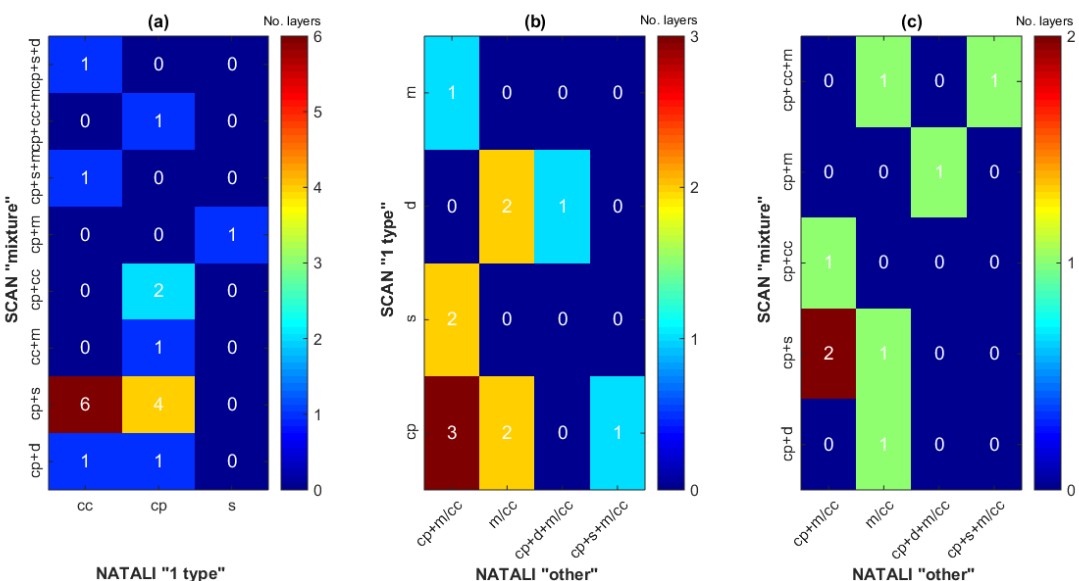

**Figure 10: Comparison between the cases classified by (a) SCAN as "mixture" and as "1 type" by NATALI, (b) SCAN as "1 type" and as "other" by NATALI, and (c) SCAN as "mixture" and as "other" by NATALI.**

From Fig. 10 it can be concluded that 12 cases that SCAN classified as "Continental Polluted and Smoke" (cp+s), NATALI classified 6 of them as "Clean Continental" (cc), 4 as "Continental Polluted" (cp) (Fig. 10a) and 2 as "Continental Polluted and Marine/Cloud Contaminated" (cp+m/cc) (Fig. 10c). Moreover, 6 cases that SCAN classified as "Continental Polluted" (cp), 3 of them were classified as "Continental Polluted and Marine or Cloud Contaminated" (cp+m/cc), 2 as "Marine/Cloud Contaminated (m/cc) and only 1 as "Continental Polluted, Smoke and Marine or Cloud Contaminated" (cp+s+m/cc) by NATALI (Fig. 10b). It seems that the aerosol optical properties of this mixture are attributed either to "Clean Continental" or to "Continental Polluted" aerosol types based on the NATALI classification.

## 3.2 Aerosol Optical Properties

The mean values of the aerosol optical properties derived from the NATALI, MD and SCAN classification for each aerosol type are presented in Table 2. The correspondence between the aerosol types and the terminology defined inside the classification methods are presented in Table 1.

**Clean Continental aerosols (cc)**

Aerosol layers classified as "Clean Continental" both by NATALI and MD present medium $lr_{355}$ values (45-46±5 sr), medium to low $lr_{532}$ values (37-39±5 sr), medium $A_{b355/532}$ and $A_{b532/1064}$ values (1.0±0.3), high $A_{e\lambda1/\lambda2}$ values (2.0±0.3) and low pldr values at 532 nm (3±1 %). These values are in accordance with others reported in previous studies concerning cc aerosols (Ansmann et al., 2001; Omar et al., 2009; Giannakaki et al., 2010).



**Continental Polluted aerosols (cp)**

Aerosol layers classified as "Continental Polluted" by both NATALI and MD present medium $lr_{355}$ nm (57±6 sr), slightly higher $lr_{532}$ values (62±7 sr), medium $A_{b355/532}$, $A_{b532/1064}$ and $A_{e355/532}$ values (1.1-1.4±0.3) and low lpdr values at 532 nm (3±1%). On the other hand the aerosol layers similarly classified by SCAN present medium $lr_{355}$ values (50±6 sr) and $lr_{532}$ values (49±5 sr), medium $A_{b355/532}$, $A_{b532/1064}$ and $A_{e355/532}$ values (1.0-1.5±0.3) and low pldr values at 532 nm (3±1%). These values are, also, in accordance with those reported in previous studies concerning this type of aerosols (Müller et al., 2007; Giannakaki et al., 2010; Gross et al., 2013; Burton et al., 2013). The similarity of these values with those of the "Clean Continental" aerosol type is the reason why it remains difficult to distinguish between these two aerosol types.

**Smoke aerosols (s)**

Smoke aerosol layers according to SCAN show medium $lr_{355}$ values (50±5 sr), medium to small $lr_{523}$ values (37±4 sr), medium $A_{b355/532}$ and $A_{b532/1064}$ values (0.9-1.3±0.3), medium to high $A_{e355/532}$ values (1.6±0.3) and low pldr values at 532 nm (3±1 %). These values are in accordance with those reported in previous studies concerning this type of aerosols (Wandinger et al., 2002; Müller et al., 2005; Müller et al., 2007; Burton et al., 2013). Again, the similarity of these values with those of the "Clean Continental" and "Continental Polluted" aerosol type is the reason why it remains difficult to distinguish between these three aerosol types.

**Marine/Cloud Contaminated aerosols (m/cc)**

Aerosol layers classified as "Marine/Cloud Contaminated" by NATALI showed low values of the $lr_{355}$ (35±4 sr), even lower $lr_{532}$ values (27±3 sr), small to medium $A_{b355/532}$ and $A_{b532/1064}$ values (0.9-1.0±0.3), increased $A_{e355/532}$ values (1.7±0.3) and low $pldr_{532}$ values (4±2 %). These values are in accordance with those reported by Cattral et al. (2005); Burton et al. (2012); Burton et al. (2013) and Gross et al. (2016), concerning the marine aerosols.

**Dust and Marine aerosols (d&m)**

Concerning the "Dust and Marine" mixture according to the MD algorithm classification, these aerosols showed medium $lr_{355}$ values (43±4 sr), low $lr_{532}$ values (46±5 sr), small $A_{b355/532}$, $A_{b532/1064}$ (0.0-0.7±0.2), $A_{e355/532}$ values (-0.2±0.2) and medium $pldr_{532}$ values (15±5 %). These values indicate large and depolarizing aerosol particles confirming the type of these particles as a mixture of dust and marine ones, according to Burton et al. (2012) and Papagiannopoulos et al. (2016a).

**Continental Polluted and Smoke aerosols (cp&s)**

The "Continental Polluted and Smoke" mixed aerosols classified according to SCAN showed medium $lr_{355}$ values (52±8 sr), medium $lr_{532}$ values (47±7 sr), medium $A_{b355/532}$ and $A_{b532/1064}$ values (0.8-1.3±0.4), high values of the $A_{e355/532}$ (1.5±0.4) and low ones of the $pldr_{532}$ (4±2 %). The medium $lr_{355}$ and $lr_{532}$ values indicate continental polluted aerosols (Müller et al., 2007;





Giannakaki et al., 2010; Gross et al., 2013; Burton et al., 2013), while the high $A_{e355/532}$ values indicate smoke aerosols

(Wandinger et al., 2002; Müller et al., 2005).

**Continental Polluted and Marine aerosols (cp&m)**

"Continental Polluted and Marine" mixture according to SCAN showed medium values of the $lr_{355}$ values (45±6 sr), increased $lr_{532}$ values (55±7 sr), low $A_{b355/532}$, $A_{b532/1064}$ and $A_{e355/532}$ values (0.3-0.7±0.3) and low $pldr_{532}$ values (8±4 %). The low pldr values are indicative for the non depolarizing aerosols such as continental polluted (Müller et al., 2007; Giannakaki

et al., 2010; Gross et al., 2013; Burton et al., 2013) and marine aerosols (Gross et al., 2011; Burton et al., 2012; Burton et al., 2013; Gross et al., 2013; Gross et al., 2016). Additionally, the low $A_{b355/532}$, $A_{b532/1064}$ and $A_{e355/532}$ values are indicative for the coarse mode aerosols such as marine, while the increased $lr_{532}$ values are more indicative for the continental polluted aerosols, rather than the marine ones.

**Continental Polluted, Dust and Marine or Clean Continental aerosols (cp&d&m/cc)**

Finally, the "Continental Polluted, Dust and Marine or Cloud Contaminated" aerosol mixture classified by NATALI showed medium $lr_{355}$ values (41±4 sr), medium $lr_{532}$ values (43±5 sr), low $A_{b355/532}$, $A_{b532/1064}$ and $A_{e355/532}$ values (-0.1-0.7±0.3), as well as medium $pldr_{532}$ values (13±4 %). Here, the medium pldr values are indicative for dust mixtures (Gross et al., 2011; Burton et al., 2013; Gross et al., 2016), while the low $A_{b355/532}$, $A_{b532/1064}$ and $A_{e355/532}$ values are indicative for the coarse mode aerosols, such as the dust and marine.

## 4 Conclusions


In this study, we compared three independent aerosol classification methods: the "Neural Network Aerosol Typing Algorithm", the "Mahalanobis distance automatic aerosol type classification" and the "Source Classification Analysis" using 97 free tropospheric aerosol layers from 4 EARLINET stations (Bucharest, Kuopio, Leipzig and Potenza) from 2014-2018. NATALI is an automated aerosol layer classification neural network, depending on the aerosol optical properties

(3β+2α+1δ) directly obtained from the EARLINET database. MD is an automated, aerosol layer classification algorithm, depending on the mean values of the aerosol optical properties ($A_{e355/1064}$, $lr_{532}$, $lr_{532}/lr_{\lambda355}$, $pldr_{532}$ and $A_{b1064/532}$) of the probed atmospheric layers. SCAN, introduced for the first time in this study, is based on the automatization of the typical classification method, while its classification procedure is based on the amount of time that an air parcel spends over specific pre-characterized aerosol source regions and a number of additional criteria, as analytically presented.

We concluded that NATALI showed the lower percentage (4%) of unclassified layers. When compared to MD, NATALI's "'X' or Cloud Contaminated" aerosol layers (where 'X' is either an aerosol type or a mixture) are classified by MD as "Clean Continental" layers, except when 'X' is a mixture of dust aerosols. When compared to SCAN, SCAN's "Continental Polluted and Smoke" layers are classified by NATALI, as either "Clean Continental" or "Continental Polluted".

Furthermore, we found that MD was unable to classify almost the 50% of the under study layers. Compared to NATALI,
these layers belong either to one single aerosol type or to aerosol mixtures. Concerning the MD's "unknown" category and
NATALI's one single aerosol types, we showed that MD's mean percentages quite well predict the aerosol type of each
layer, even though this aerosol type is not chosen by the classification process of MD. Concerning the MD's "unknown" and
NATALI's "mixtures" categories, MD algorithm revealed an increased contribution of dust and dust aerosols mixtures
(approximately 50%) inside the studied aerosol layers. Compared to SCAN, MD's "unknown" layers are, mainly, either
"Continental Polluted" or "Continental Polluted and Smoke". Finally, SCAN's "Continental Polluted and Smoke" layers are
classified by MD either as "Clean Continental" or "Continental Polluted".

We found that the SCAN code successfully managed to classify more than 50% of the layers studied, either as a single
aerosol type or as mixtures of different aerosols. Being aerosol optical property independent, SCAN provides the advantage
that its classification process is not affected by the overlapping values of the optical properties representing more than one
aerosol types (ex. clean continental, continental polluted, smoke). Furthermore, it has no limitations concerning its ability to
classify aerosol mixtures, an advantage that occurs from the air mass trajectory analysis and the relevant aerosol sources at
ground. Finally, it can be useful for all types of lidar systems (independently of the number of channels used), as well as for
other network-based systems (radar profilers, sunphotometers etc.).

**Author contribution**

Doina Nicolae, Nikolaos Papagiannopoulos and Elina Giannakaki distributed the NATALI, MD and SCAN algorithms,
respectively. Christinna-Anna Papanikolaou created algorithms which produce the maps presented in this work. Elina
Giannakaki had the idea of this paper. Maria Mylonaki upgraded the SCAN algorithm, collected the lidar data, made the
comparison, analyzed the results and wrote the paper. All authors participated in scientific discussions on this study and
reviewed/edited the manuscript during its preparation process.

**Competing interests**

"The authors declare that they have no conflict of interest."

**Acknowledgments**

The research work was supported by the Hellenic Foundation for Research and Innovation (HFRI) under the HFRI PhD
Fellowship grant (Fellowship Number: 669). We acknowledge support of this work by the project "PANhellenic
infrastructure for Atmospheric Composition and climatE change" (MIS 5021516) which is implemented under the Action
"Reinforcement of the Research and Innovation Infrastructure", funded by the Operational Programme "Competitiveness,


Entrepreneurship and Innovation" (NSRF 2014-2020) and co-financed by Greece and the European Union (European Regional Development Fund). The authors acknowledge support through ACTRIS under grand agreement no. 262 254 of the European Commission Seventh Framework Programme (FP7/2007-2013) and ACTRIS-2 under grant agreement no. 654109

from Horizon 2020 research and innovation program of the European Commission. The authors gratefully acknowledge the NOAA Air Resources Laboratory (ARL) for the provision of the HYSPLIT transport and dispersion model and/or READY website (http://www.ready.noaa.gov) used in this publication. We acknowledge the use of data products or imagery from the Land, Atmosphere Near real-time Capability for EOS (LANCE) system operated by NASA's Earth Science Data and Information System (ESDIS) with funding provided by NASA Headquarters.

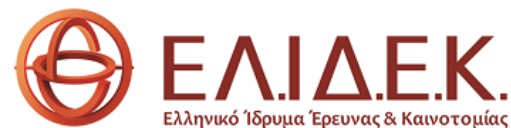


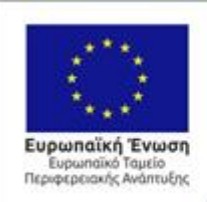 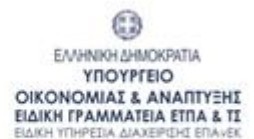 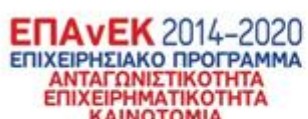 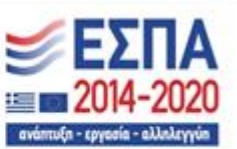

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





**Table 1: Correspondence between the aerosol types and shorthand used for this work and the actual aerosol types defined inside the NATALI, MD and SCAN aerosol classification algorithms.**

| Aerosol type | NATALI | MD | SCAN |
|---|---|---|---|
| Continental (cc) | Continental | Continental | Clean Continental |
| Continental Polluted (cp) | Continental polluted | Continental polluted | Continental polluted |
| Smoke (s) | Smoke | Smoke | Smoke |
| Dust (d) | Dust | Dust | Dust |
| Marine (m) | Marine | Marine | Marine |
| Volcanic (v) | Volcanic | Volcanic | Volcanic |
| Continental and dust (cp+d) | Continental dust | Dust polluted | Continental and dust |
| Dust and marine (d+m) | Marine mineral | Mixed dust | Dust and marine |
| Continental and smoke (cp+s) | Continental smoke | - | Continental polluted and smoke |
| Dust and smoke (d+s) | Dust polluted | Dust polluted | Dust and smoke |
| Continental and marine (cc+m) | Coastal | - | Clean continental and marine |
| Continental polluted and marine (cp+m) | Coastal polluted | - | Continental polluted and marine |
| Continental polluted and clean continental (cp+cc) | - | - | Continental polluted and clean continental |
| Continental and dust and marine (cp+d+m) | Mixed dust | - | Continental polluted and dust and marine |
| Continental and smoke and marine (cc+s+m) | - | - | Clean continental and smoke and marine |
| Continental polluted and smoke and marine (cp+s+m) | Mixed smoke | - | Continental polluted and smoke and marine |





| | | | |
|---|---|---|---|
| Continental and smoke and dust (cp+s+d) | - | - | Continental and smoke and dust |
| Continental and clean continental and marine (cp+cc+m) | - | - | Continental and clean continental and marine |




**Table 2: EARLINET lidar station information**

| Location | ACTRIS Code | Institute | Coordinates (lat, lon, altitude amsl.) | Reference | No of layers | Selected period |
|---|---|---|---|---|---|---|
| Bucharest | INO | National Institute of R&D for Optoelectronics (INOE) | 44.35 N, 26.03 E, 93 m | Nemuc, et al. 2013 | 7 | 2017 |
| Kuopio | KUO | Finnish Meteorological Institute (FMI), Atmospheric Research Centre of Eastern Finland, Kuopio | 62.74 N, 27.54 E, 190 m | Althausen, et al., 2009, Engelmann, et al., 2016 | 9 | 2015, 2016 |
| Leipzig | LEI | Leibniz Institute for Tropospheric Research, Leipzig | 51.35 N, 12.43 E, 90 m | Althausen, et al., 2009, Engelmann, et al., 2016 | 17 | 2018 |
| Potenza | POT | Consiglio Nazionale delle Ricerche - Istituto di Metodologie per l'Analisi Ambientale (CNR-IMAA), Potenza | 40.60 N, 15.72 E, 760 m | Madonna, et al., 2011 | 64 | 2015-2016 |






**Table 3: Mean values and standard deviations of aerosol optical properties according to classification each classification method.**

| Aerosol types | Method | Clean Cont. | Cont. Polluted | Smoke | Marine/ Cl. Cont. | Dust + Marine | Cont. Polluted + Smoke | Cont. Polluted + Marine | Cont. Polluted + Dust + Marine / Cl. Cont. |
|---|---|---|---|---|---|---|---|---|---|
| No of cases | NAT | 24 | 24 | - | 11 | - | - | 14 | 7 |
| | MD | 29 | 13 | - | - | 4 | - | - | - |
| | SCAN | - | 22 | 5 | - | - | 16 | 4 | - |
| $LR_{355}$ [sr] | NAT | 46.3±5.0 | 57.5±6.0 | - | 34.6±3.5 | - | - | 69.0±11.0 | 41.4±4.3 |
| | MD | 44.9±5.1 | 57.0±6.4 | - | - | 42.5±4.4 | - | - | - |
| | SCAN | - | 50.2±5.5 | 45.8±4.7 | - | - | 52.4 ± 7.9 | 45.3±5.9 | - |
| $LR_{532}$ [sr] | NAT | 37.3±3.7 | 61.6±6.7 | - | 27.3±3.4 | - | - | 31.0±5.3 | 43.1±4.6 |
| | MD | 38.9±4.6 | 61.0±6.9 | - | - | 46.0±4.7 | - | - | - |
| | SCAN | - | 49.2±5.4 | 37.2±4.0 | - | - | 47.3 ± 7.1 | 54.8±7.2 | - |
| $A_{e355/532}$ | NAT | 2.0±0.3 | 1.2±0.3 | - | 1.7±0.3 | - | - | 0.9±0.4 | -0.1±0.3 |
| | MD | 1.6±0.3 | 1.1±0.3 | - | - | -0.2±0.2 | - | - | - |
| | SCAN | - | 1.5±0.3 | 1.6±0.3 | - | - | 1.5 ± 0.4 | 0.3±0.3 | - |
| $A_{b355/532}$ | NAT | 1.1±0.3 | 1.4±0.3 | - | 0.9±0.3 | - | - | -1.2±0.4 | 0.0±0.3 |
| | MD | 1.1±0.3 | 1.2±0.3 | - | - | 0.0±0.2 | - | - | - |
| | SCAN | - | 1.2±0.3 | 0.9±0.3 | - | - | 0.8 ± 0.4 | 0.5±0.3 | - |
| $A_{b532/1064}$ | NAT | 1.2±0.2 | 1.1±0.2 | - | 1.0±0.2 | - | - | 1.3±0.2 | 0.7±0.1 |
| | MD | 1.1±0.2 | 1.1±0.2 | - | - | 0.7±0.1 | - | - | - |
| | SCAN | - | 1.0±0.2 | 1.3±0.2 | - | - | 1.3 ± 0.2 | 0.7±0.2 | - |
| pldr [%] | NAT | 3.4±1.4 | 2.3±0.7 | - | 4.1±1.6 | - | - | 2.7±1.6 | 13.0±4.4 |
| | MD | 3.0±1.2 | 2.7±1.0 | - | - | 15.2±5.3 | - | - | - |
| | SCAN | - | 3.3±1.3 | 2.7±1.1 | - | - | 4.0 ± 1.9 | 7.7±3.8 | - |

