# Peer review of "Aerosol type classification analysis using EARLINET multiwavelength and depolarization lidar observations"

_Atmospheric Chemistry and Physics, 2020_

## Referee Comment (RC1) · Anonymous Referee #1 · 14 Nov 2020

The manuscript compares different schemes of aerosol classification. In particular, it compares aerosol types derived from multiwavelength lidar measurements (for predefined aerosol types) with new algorithm, based on the back trajectory analysis. Aerosol types from different schemes demonstrate reasonable agreement and this is important result. Manuscript is clearly written and suits for ACP. I have just technical comments.

Ln 35 – 42. I think it can be skipped or shortened. Well known things. Ln.65 By this point reader may have feeling that intensive properties for different aerosol types are well known. In reality these present strong variations even for single aerosol type. And not all of these are equally trustable. For example, backscattering Angstrom

exponent is very sensitive to variation of particle size and refractive index. I think a short paragraph, explaining complexity of the problem of aerosol classification would be useful. Fig.3a. $\beta 355$ is not zero at 2.8 km, while $\beta 532$ and $\beta 1064$ are. Is choice of reference point correct? Fig.3b No reason to show particle depolarization above 2.6 km, because results are already untrustable. Fig.3 second row. 3a. The same question about reference point at 7 km. 3.e. No reason to show particle depolarization for low backscattering (above $\sim$6.5 km). It is not trustable. Ln 185-195. Too many numbers. Many be better put them in Table? Ln 371. "Smoke aerosol layers according to SCAN show medium lr355 values (50$\pm$5 sr), medium to small lr523 values (37$\pm$4 sr)," Actually lidar ratios for smoke present very large variation. In particular, for aged smoke lr355<lr532. It depends also on humidity... Ln.384. Variability of dust lidar ratio is discussed in recent publication: Veselovskii, I., Hu, Q., Goloub, P., Podvin, T., Korenskiy, M., Derimian, E., Legrand, M., Castellanos, P.: Variability of Lidar-Derived Particle Properties Over West Africa Due to Changes in Absorption: Towards an Understanding. Atm. Chem. Phys., 20, 6563-6581, 2020.

Please also note the supplement to this comment:
https://acp.copernicus.org/preprints/acp-2020-865/acp-2020-865-RC1-supplement.pdf

---

## Referee Comment (RC2) · Anonymous Referee #2 · 25 Nov 2020

This paper reported a new algorithm to classify the aerosols layers type in a measurement site. The authors shows a comparison between two previous reported algorithms ("Neural Network Aerosol Typing Algorithm", and "Mahalanobis distance automatic aerosol type classification" and the new one ("Source Classification Analysis") using 97 free tropospheric aerosol layers from 4 EARLINET stations. Although there are some aerosol classification algorithms, this new method has the advantage that it does not need the properties of the aerosols to get the aerosol type. This report brings an important contribution to the scientific community and I believe it can be published in ACP. I have some technical comments that should be checked before publication.

[Figure]

Technical Corrections Recommended to acp-2020-865-manuscript-version1.pdf.

Ln 28: "...while MD has the percentage of unclassified..." Suggestion: "...while MD has the higher percentage of unclassified..."

Ln 38: "(Weitcamp et al., 2005)." Note: I did not find it at the references list

Ln 43 - 45: Suggestion: Please check this sentence.

Ln 111: "whie'" Suggestion: While,

Ln 116: Kavoudou et al. (2019) Note: I did not find it at the references list

Ln 168: In total, 97 FT aerosol Suggestion: This was first mentioned in the abstract, I recommend you to introduce here in the text for the first time

Ln 173 - 174: the units in the Y axis for altitude in the graphics are reported in meters, and it should be in km, as is stated in the text.

Ln 186: "Ab355/532=1.36Âň±0.0.05" Suggestion: 0.05 ???

Ln 237 - 238: Same as previous comment (Ln 173 - 174) about units of altitude in meter or kilometer?

Ln 239: (i) Kuopio and (ii) Potenza at 3000 m amsl. Note: The are not the notation i and ii in the graphics

Ln 353: "Table 2." Suggestion: Table 3 ?????

Ln 369: " lr 523" Suggestion: 532?

Ln 378: Cattral et al. (2005); Note: is it on the reference list?

Ln 379: Gross et al. (2016), Note: is it on the reference list?

Ln 384: Papagiannopoulos et al. (2016a) Note: There is only one Papagiannopoulos 2016, why a?

Ln 391: Continental Polluted and Marine aerosols (cp&m) Note: What about aerosol properties values derived from NATALI?

Ln 457: References Note: There are some papers are not referenced in the text. Please check it. eg: Amiridis et al. 2009 is not referenced in the text Ansmann et al. 2003 is not referenced in the text

Ln 786: Table 3: Mean values and standard deviations of aerosol optical properties according to classification each classification method. Note: This table is not discussed in the text.

---

## Referee Comment (RC3) · Anonymous Referee #3 · 26 Nov 2020

General

The authors present SCAN, a new classification algorithm that relies on aerosol geometrical properties from lidar measurements and the Lagrangian trajectory model HYSPLIT. They compare their algorithm with well tested aerosol classification approaches (Mahalanobis Distance, Neural Network) that are based on the aerosol intensive optical properties. The subject is interesting as this algorithm shows that, while manual classification with trajectory is a rather common approach, automated classification is also possible using geometrical aerosol information that most aerosol oriented lidar systems (and maybe even Ceilometers) can provide. Therefore, I recommend this manuscript

for publication in ACP after some minor revisions/improvements are performed.

Comments

– Section 2.1 (Source classification analysis): An altitude threshold of the trajectory is applied per domain in order for the algorithm to decide whether emitted particles could be carried away. The time the air mass remains below this altitude limit is crucial though. Is it meaningful to also include such an additional threshold in the algorithm?

– Section 2.1 (Source classification analysis): Is a single hot spot sufficient to mark a layer as smoke mixture? The authors could fine tune this limit (number of hot spots near the trajectory within a polluted/clean continental box" looking at the performance of SCAN in comparison with the other two algorithms

– Section 2.1 (Source classification analysis): The authors should add a few lines here to explain the link between the aerosol type and the 4 kinds of domains. While this is straightforward for marine and dust types, this is not the case for the continental polluted aerosols. The polluted continental domain on the map probably correspond to regions with increased urban activity but this has to be specified in the manuscript.

– Section 2.1 (Neural network aerosol classification algorithm): Have the authors checked the performance of NATALI by applying different user configurations? Pieces of information such as the selected confidence level and minimum agreement threshold should be provided here.

– Section 2.3 (Case studies): A lot of numbers are provided in this part which makes it hard to follow. It would be more efficient if they were presented in tables.

– Line 111: Please specify also the accumulated probability that corresponds to a MD of 4.3 with 4 degree of freedom (independent variables). In addition, the authors explain why these thresholds are applied. Are they supported by previous studies?

– Line 168: This technique is also applied in NATALI. Is the layering automatically performed by SCAN or is the analysis based on layers obtained by NATALI? In the

latter case, it has to be specified in the manuscript (e.g. "Source classification analysis" Section) that only the classification part of SCAN is automated, and not the layering part.

– Lines 224-251: Care has to be taken here because the Mahalanobis distance algorithm utilizes both two different probability metrics, the chi-squared probability (Mahalanobis Distance) and also the normalized probability. Which one are the authors referring to?

Minor Corrections

– Line 28: "...while MD has the percentage of..." Do the authors mean highest percentage?

– Line 55: Please correct the typo "neither objective nor automated"

– Line 117: Please replace "3+2" with "3b_$\lambda\alpha$+2a_$\lambda\alpha$"

– Figure 3: I suggest that the authors use different colors for the layer base and top. Otherwise the layer boundaries can be confusing for the Extinction and Backscatter profiles.

– Line 423: "dust and dust aerosols mixtures" Is this a typo?

– Figure 5: Please check the percentages of pie charts of Figure 5 as the do not sum up to exactly 100%.

– Figure 6: Please check the percentages of pie charts of Figure 6 as the do not sum up to exactly 100%.

---

## Author Comment (AC1) · 26 Dec 2020

Author's response to reviewers for the paper submitted to Atmospheric Chemistry and Physics entitled: "Aerosol type classification analysis using EARLINET multiwavelength and depolarization lidar observations".

**Response to Reviewer #1:**

The authors would like to thank the reviewer for carefully reading the manuscript and for providing constructive remarks to improve this paper. The replies to the comments are given in the paragraphs below. The referee comments are highlighted in blue, while our responses in black fonts. The text modifications in the revised manuscript are highlighted in red.

Ln 35 – 42. I think it can be skipped or shortened. Well known things.

These paragraphs were slightly modified.

Initial: "Furthermore, aerosols affect cloud formation and behaviour both serving as seeds (cloud condensation nuclei, ice nuclei) upon which cloud droplets and ice crystals form, and influencing the cloud albedo due to changing concentrations of cloud condensation and ice nuclei, also known as the Twomey effect (aerosol-cloud interaction, "aci") (Twomey, 1959; Twomey and Warner, 1967; Hobbs et al., 1993; Stevens and Feigold, 2009; IPCC, 2014; Rosenfeld et al., 2014; Rosenfeld et al., 2016).

One accurate and powerful technique to study atmospheric aerosols is the light detection and ranging (lidar) which is based on the active remote sensing of the atmosphere (Weitcamp et al., 2005). This technique has received quite an attention, because of the multiple possibilities to retrieve near real-time information of the structure and the composition of the atmosphere with high spatial (i.e. down to few meters) and temporal (i.e. down to seconds depending on the system) resolution."

Changed: "Furthermore, aerosols affect cloud formation and behavior, not only serving as seeds (cloud condensation nuclei, ice nuclei) upon which cloud droplets and ice crystals form, but also influencing the cloud albedo due to changing concentrations of cloud condensation and ice nuclei, also known as the Twomey effect (Twomey, 1959; IPCC, 2014; Rosenfeld et al., 2014, 2016).

The light detection and ranging (lidar) technique which is based on the active remote sensing of the atmosphere (Weitcamp et al., 2005) has received quite an attention, because of the multiple possibilities to retrieve near real-time information of the vertical structure and the composition of the atmosphere with high spatial (i.e. down to few meters) and temporal (i.e. down to seconds depending on the system) resolution."

Ln.65 By this point reader may have feeling that intensive properties for different aerosol types are well known. In reality these present strong variations even for single aerosol type. And not all of these are equally trustable. For example, backscattering Angstrom exponent is very sensitive to variation of particle size and refractive index. I think a short paragraph, explaining complexity of the problem of aerosol classification would be useful.

We added a few lines where we present the complexity of the aerosol classification which depends on the intensive aerosol optical properties.

"In reality, the intensive aerosol optical properties can vary greatly even for single aerosol types. For example, Nicolae et al. (2013) showed that fresh biomass burning aerosols have

higher Ångström exponents and refractive indexes than the aged ones. Additionally, according to Veselovkii et al. (2020), the lidar ratio values of dust aerosols can vary greatly depending on the source region minerology. Thus, the physico-chemical modifications which the aerosols undergo, from the time they are created to when they are finally observed, change their geometrical, size and optical characteristics and as a result, their optical properties as well."

Fig.3a. β355 is not zero at 2.8 km, while β532 and β1064 are. Is choice of reference point correct?

All lidar data used were marked as QC2 from the Quality Control procedure of EARLINET. Please, find more information about the Quality Control procedure of EARLINET in https://www.earlinet.org/index.php?id=293.

Fig.3b No reason to show particle depolarization above 2.6 km, because results are already untrustable.

This Figure was modified as suggested.

[Figure]

Fig.3 second row. 3a. The same question about reference point at 7 km.

All lidar data used were marked as QC2 from the Quality Control procedure of EARLINET. Please, find more information about the Quality Control procedure of EARLINET in https://www.earlinet.org/index.php?id=293.

3.e. No reason to show particle depolarization for low backscattering (above ~6.5 km). It is not trustable.

This Figure was modified as suggested.

[Figure]

Ln 185-195. Too many numbers. Many be better put them in Table?

We added a Table.

Initial: "The mean values of the intensive aerosol optical properties within the aerosol layer observed over Kuopio are: $lr_{355}$=65.58±11.02 sr, $lr_{532}$=72.51±17.61 sr, $A_{e355/532}$=1.23±0.62, $A_{b355/532}$=1.36±0.0.05, $A_{b532/1064}$=1.23±0.05 and $pldr_{532}$=2.1±0.1 %, indicating fine absorbing aerosols. The mean values of the intensive aerosol optical property within the lower aerosol layer observed over Potenza on the same day are: $lr_{355}$=35.97±1.09 sr, $lr_{532}$=24.55±4.15 sr, $A_{e355/532}$=1.14±0.44, $A_{b355/532}$=0.16±0.04, $A_{b532/1064}$=0.97±0.04, and $pldr_{532}$=13.5±0.4 %. Similarly, for the middle aerosol layer observed over Potenza: $lr_{355}$=31.05±2.11 sr, $lr_{532}$=22.50±1.69 sr, $A_{e355/532}$=0.86±0.25, $A_{b355/532}$=0.06±0.13, $A_{b532/1064}$=0.79±0.06 and $pldr_{532}$=15.4±1.5 %. These values of both lower and middle aerosol layers are indicative of coarse semi-depolarizing aerosols, probably, mixed dust particles. Finally, for the upper aerosol layer observed over Potenza: $lr_{355}$=38.77±4.81 sr, $lr_{532}$=24.44±3.39 sr, $A_{e355/532}$=0.56±0.37, $A_{b355/532}$=-0.58±0.25, $A_{b532/1064}$=0.72±0.07 and $pldr_{532}$=24.8±1.0 %, indicating coarse high-depolarizing aerosols, probably of dust origin."

Changed: "The mean values of the intensive aerosol optical properties within the aerosol layer observed over Kuopio and Potenza on 30 July 2015 are also presented in Table 3. In particular, for the case of Kuopio, the $lr_{355}$ values were found to be lower than those of $lr_{532}$, the Ångström exponent (both extinction- and backscatter-related) was higher than 1.2 and pldr532 had low values (<5 %), indicating fine absorbing aerosols. For the case of Potenza, the lr values were found to be low (<39 sr at 355 nm and <25 sr at 532 nm), while the Ångström exponents remained mainly below 1.0 for all the three aerosol layers observed. The difference between these three aerosol layers is the value of $pldr_{532}$ which was found to be ascended from 13.5±0.4 % (bottom layer) to 15.4±1.5 % (middle layer) and finally reached the value of 24.8±1.0 % (top layer), indicating coarse semi-depolarizing aerosols at lower altitudes (<4.5 km) and high-depolarizing aerosols higher, probably of dust origin.

Table 3: Mean values of the intensive optical properties of aerosol layers observed on 30 July 2015 over Kuopio and Potenza.

| Site | Height [km] | $lr_{355}$ [sr] | $lr_{532}$ [sr] | $A_{e355/532}$ | $A_{b355/532}$ | $A_{b532/1064}$ | $pldr_{532}$ [%] |
|---|---|---|---|---|---|---|---|
| Kuopio | 1.5-1.9 | 65.58±11.02 | 72.51±17.61 | 1.23±0.62 | 1.36±0.05 | 1.23±0.05 | 2.1±0.1 |

| | | | | | | | |
|---|---|---|---|---|---|---|---|
| Potenza (bottom) | 2.8-3.1 | 35.97±1.09 | 24.55±4.15 | 1.14±0.44 | 0.16±0.04 | 0.97±0.04 | 13.5±0.4 |
| Potenza (middle) | 3.4-3.9 | 31.05±2.11 | 22.50±1.69 | 0.86±0.25 | 0.06±0.13 | 0.79±0.06 | 15.4±1.5 |
| Potenza (top) | 4.5-5.4 | 38.77±4.81 | 24.44±3.39 | 0.56±0.37 | -0.58±0.25 | 0.72±0.07 | 24.8±1.0 |

"

Ln 371. "Smoke aerosol layers according to SCAN show medium lr355 values (50±5 sr), medium to small lr523 values (37±4 sr)," Actually lidar ratios for smoke present very large variation. In particular, for aged smoke lr355<lr532. It depends also on humidity. . .

This aspect is now presented at the Introduction section of this manuscript.

"In reality, the intensive aerosol optical properties can vary greatly even for single aerosol types. For example, Nicolae et al. (2013) showed that fresh biomass burning aerosols have higher Ångström exponents and refractive indexes than the aged ones. Additionally, according to Veselovkii et al. (2020), the lidar ratio values of dust aerosols can vary greatly depending on the source region minerology. Thus, the physico-chemical modifications which the aerosols undergo, from the time they are created to when they are finally observed, change their geometrical, size and optical characteristics and as a result, their optical properties as well."

Ln.384. Variability of dust lidar ratio is discussed in recent publication: Veselovskii, I., Hu, Q., Goloub, P., Podvin, T., Korenskiy, M., Derimian, E., Legrand, M., Castellanos, P.: Variability of Lidar-Derived Particle Properties Over West Africa Due to Changes in Absorption: Towards an Understanding. Atm. Chem. Phys., 20, 6563-6581, 2020.

The Veselovskii et al. 2020 paper deals with dust cases having pldr>30 % and a variety of LR values. They observed the presence of different dust types with other ways rather than the criteria of the pldr value being higher than a certain threshold, which was not sufficient to make the separation between different types of dust. They concluded that the observed variations in $S_{355}/S_{532}$ can be related to the source region mineralogy. At this point of our manuscript, dust with marine mixtures were studied having pldr values ~15 %. However, this study is now referenced in the introduction section.

Added: "In reality, the intensive aerosol optical properties can vary greatly even for single aerosol types. For example, Nicolae et al. (2013) showed that fresh biomass burning aerosols have higher Ångström exponents and refractive indexes than the aged ones. Additionally, according to Veselovkii et al. (2020), the lidar ratio values of dust aerosols can vary greatly depending on the source region minerology. Thus, the physico-chemical modifications which the aerosols undergo, from the time they are created to when they are finally observed, change their geometrical, size and optical characteristics and as a result, their optical properties as well."

**Response to Reviewer #2:**

The authors would like to thank the reviewer for carefully reading the manuscript and for providing constructive remarks to improve this paper. The replies to the comments are given in the paragraphs below. The referee comments are highlighted in blue, while our responses in black fonts. The text modifications in the revised manuscript are highlighted in red.

"Ln 28: ". . .while MD has the percentage of unclassified. . ." Suggestion: ". . .while MD has the higher percentage of unclassified. . .""

The word "higher" was added at the sentence.

Initial: "Finally, our results show that NATALI has the lower percentage of unclassified layers (4%), while MD has the percentage of unclassified layers (50%) and the lower percentage of cases classified as aerosol mixtures (5%)."

Changed: "Finally, our results show that NATALI has the lower percentage of unclassified layers (4%), while MD has the higher percentage of unclassified layers (50%) and the lower percentage of cases classified as aerosol mixtures (5%)."

"Ln 38: "(Weitcamp et al., 2005)." Note: I did not find it at the references list"

Weitcamp et al., 2005 has been added at the reference list.

"Weitkamp, C., (Eds.): Lidar, Springer-Verlag, New York, 2005."

"Ln 43 - 45: Suggestion: Please check this sentence." Να τα σπάσω σε 3 προτάσεις.

The authors modified this part.

Initial: "Specifically, the multi-wavelength Raman/depolarization lidars can be used for aerosol detection and characterization (i.e. dust, smoke, continental, etc.) as they provide vertically-resolved information of extensive [particle backscatter ($b_{\lambda\alpha}$) and extinction coefficients ($e_{\lambda\alpha}$)] and intensive aerosols properties [lidar ratio ($lr_{\lambda\alpha}$), Ångström exponent extinction- ($Ae_{\lambda\alpha/\lambda\beta}$) and backscatter-related ($Ab_{\lambda\alpha/\lambda\beta}$), particle linear depolarization ratio (pldr)] optical properties (Nicolae et al., 2006; Burton et al., 2012; Groß et al., 2013; Giannakaki et al., 2016; Soupiona et al., 2019)."

Changed: "Specifically, the multi-wavelength Raman/depolarization lidars can be used for aerosol detection and characterization (i.e. dust, smoke, continental, etc.). They can provide vertically-resolved information of the extensive and intensive aerosol optical properties (Nicolae et al., 2006; Burton et al., 2012; Groß et al., 2013; Giannakaki et al., 2016; Soupiona et al., 2019). These properties are the particle backscatter ($b_{\lambda\alpha}$) and extinction coefficients ($e_{\lambda\alpha}$), the lidar ratio ($lr_{\lambda\alpha}$), Ångström exponent extinction- ($Ae_{\lambda\alpha/\lambda\beta}$) and backscatter-related ($Ab_{\lambda\alpha/\lambda\beta}$) and the particle linear depolarization ratio (pldr)."

"Ln 111: "whieâAŽ" Suggestion: While,"

The word "while" was corrected and a comma was added.

"In this study, we used four aerosol intensive properties: the backscatter-related Ångström exponent at 355 and 1064 nm, the aerosol lidar ratio at 532 nm, the color ratio of the lidar ratios and the aerosol particle linear depolarization ratio at 532 nm, while, the minimum accepted distance was set to 4.3."

"Ln 116: Kavoudou et al. (2019) Note: I did not find it at the references list"

The Kavoudou et al. (2019) was replaced by Voudouri et al., (2019) and the referenced was added at the reference list.

"Voudouri, K. A., Siomos, N., Michailidis, K., Papagiannopoulos, N., Mona, L., Cornacchia, C., Nicolae, D., and Balis, D.: Comparison of two automated aerosol typing methods and their application to an EARLINET station, Atmos. Chem. Phys., 19, 10961–10980, doi:10.5194/acp-19-10961-2019, 2019."

"Ln 168: In total, 97 FT aerosol Suggestion: This was first mentioned in the abstract, I recommend you to introduce here in the text for the first time"

This sentence was modified.

Initial: "In total, 97 FT aerosol layers were obtained and their mean aerosol optical properties (intensive and extensive) were calculated."

Changed: "In total, 97 free tropospheric (FT) aerosol layers were obtained and their mean aerosol optical properties (intensive and extensive) were calculated."

"Ln 173 - 174: the units in the Y axis for altitude in the graphics are reported in meters, and it should be in km, as is stated in the text."

The figures were improved and now the altitude is provided in kilometers.

[Figure]

[Figure]

"Ln 186: "Ab355/532=1.36Ǎn̆±0.0.05" Suggestion: 0.05 ???"

We deleted the numbers from the text and put them in a new Table.

Table 3: Mean values of the intensive optical properties of aerosol layers observed on 30 July 2015 over Kuopio and Potenza.

| Site | Height [km] | $lr_{355}$ [sr] | $lr_{532}$ [sr] | $A_{e355/532}$ | $A_{b355/532}$ | $A_{b532/1064}$ | $pldr_{532}$ [%] |
|---|---|---|---|---|---|---|---|
| Kuopio | 1.5-1.9 | 65.58±11.02 | 72.51±17.61 | 1.23±0.62 | 1.36±0.05 | 1.23±0.05 | 2.1±0.1 |
| Potenza (bottom) | 2.8-3.1 | 35.97±1.09 | 24.55±4.15 | 1.14±0.44 | 0.16±0.04 | 0.97±0.04 | 13.5±0.4 |
| Potenza (middle) | 3.4-3.9 | 31.05±2.11 | 22.50±1.69 | 0.86±0.25 | 0.06±0.13 | 0.79±0.06 | 15.4±1.5 |
| Potenza (top) | 4.5-5.4 | 38.77±4.81 | 24.44±3.39 | 0.56±0.37 | -0.58±0.25 | 0.72±0.07 | 24.8±1.0 |

"Ln 237 - 238: Same as previous comment (Ln 173 - 174) about units of altitude in meter or kilometer?" and "Ln 239: (i) Kuopio and (ii) Potenza at 3000 m amsl. Note: The are not the notation i and ii in the graphics"

The figure was improved.

[Figure]

The numbering of the tables was changed because a new table was added. "Table 2" was replaced by "Table 4".

Initial: "The mean values of the aerosol optical properties derived from the NATALI, MD and SCAN classification for each aerosol type are presented in Table 2."

Changed: "The mean values of the aerosol optical properties derived from the NATALI, MD and SCAN classification for each aerosol type are presented in Table 4 and discussed in this section."

We deleted the numbers from the text and put them in a new Table.

Table 3: Mean values of the intensive optical properties of aerosol layers observed on 30 July 2015 over Kuopio and Potenza.

| Site | Height [km] | $lr_{355}$ [sr] | $lr_{532}$ [sr] | $A_{e355/532}$ | $A_{b355/532}$ | $A_{b532/1064}$ | $pldr_{532}$ [%] |
|---|---|---|---|---|---|---|---|
| Kuopio | 1.5-1.9 | 65.58±11.02 | 72.51±17.61 | 1.23±0.62 | 1.36±0.05 | 1.23±0.05 | 2.1±0.1 |
| Potenza (bottom) | 2.8-3.1 | 35.97±1.09 | 24.55±4.15 | 1.14±0.44 | 0.16±0.04 | 0.97±0.04 | 13.5±0.4 |
| Potenza (middle) | 3.4-3.9 | 31.05±2.11 | 22.50±1.69 | 0.86±0.25 | 0.06±0.13 | 0.79±0.06 | 15.4±1.5 |
| Potenza (top) | 4.5-5.4 | 38.77±4.81 | 24.44±3.39 | 0.56±0.37 | -0.58±0.25 | 0.72±0.07 | 24.8±1.0 |

"Ln 378: Cattral et al. (2005); Note: is it on the reference list?"

The reference was added at the reference list.

"Cattrall, C., Reagan J., Thome K., and Dubovik O.: Variability of aerosol and spectral lidar and backscatter andextinction ratios of key aerosol types derived from selected Aerosol Robotic Network locations, J. Geophys. Res., 11, D10S11, doi:10.1029/2004JD005124, 2005."

"Ln 379: Gross et al. (2016), Note: is it on the reference list?"

This reference was deleted from the text.

"Ln 384: Papagiannopoulos et al. (2016a) Note: There is only one Papagiannopoulos 2016, why a?"

The "a" was deleted.

"Papagiannopoulos et al. (2016)."

"Ln 391: Continental Polluted and Marine aerosols (cp&m) Note: What about aerosol properties values derived from NATALI?"

This paragraph was improved.

Initial: ""Continental Polluted and Marine" mixture according to SCAN showed medium values of the $lr_{355}$ values (45±6 sr), increased $lr_{532}$ values (55±7 sr), low $A_{b355/532}$, $A_{b532/1064}$ and $A_{e355/532}$ values (0.3-0.7±0.3) and low $pldr_{532}$ values (8±4 %). The low pldr values are indicative for the non-depolarizing aerosols such as continental polluted (Müller et al., 2007; Giannakaki et al., 2010; Gross et al., 2013; Burton et al., 2013) and marine aerosols (Gross et al., 2011; Burton et al., 2012; Burton et al., 2013; Gross et al., 2013; Gross et al., 2016). Additionally, the low $A_{b355/532}$, $A_{b532/1064}$ and $A_{e355/532}$ values are indicative for the coarse mode aerosols such as

marine, while the increased lr$_{532}$ values are more indicative for the continental polluted aerosols, rather than the marine ones."

Changed: ""Continental Polluted and Marine" mixture according to NATALI showed a large difference between the lr355 and lr532 values with the latter being smaller (lr355=69±11 sr, lr532=31±5.3 sr). The Ae355/532, Ab355/532 and pldr at 532 nm showed low values (0.9±0.4, -1.2±0.4, 2.7±1.6 respectively), while the Ab532/1064 showed large values (1.3±0.2). According to SCAN showed medium values of the lr355 values (45±6 sr), increased lr532 values (55±7 sr), low Ab355/532, Ab532/1064 and Ae355/532 values (0.3-0.7±0.3) and low pldr532 values (8±4 %). The low pldr values are indicative for the non depolarizing aerosols such as continental polluted (Müller et al., 2007; Giannakaki et al., 2010; Gross et al., 2013; Burton et al., 2013) and marine aerosols (Gross et al., 2011; Burton et al., 2012; Burton et al., 2013; Gross et al., 2013). Additionally, the low Ab355/532, Ab532/1064 and Ae355/532 values are indicative for the coarse mode aerosols such as marine, while the increased lr532 values are more indicative for the continental polluted aerosols, rather than the marine ones."

"Ln 457: References Note: There are some papers are not referenced in the text. Please check it. eg: Amiridis et al. 2009 is not referenced in the text Ansmann et al. 2003 is not referenced in the text"

These two references were deleted from the reference list.

"Ln 786: Table 3: Mean values and standard deviations of aerosol optical properties according to classification each classification method. Note: This table is not discussed in the text."

It is discussed in section 3.2. The numbering of the tables was not right.

Initial: "The mean values of the aerosol optical properties derived from the NATALI, MD and SCAN classification for each aerosol type are presented in Table 2. The correspondence between the aerosol types and the terminology defined inside the classification methods are presented in Table 1."

Changed: "The mean values of the aerosol optical properties derived from the NATALI, MD and SCAN classification for each aerosol type are presented in Table 3 and discussed in this section. The correspondence between the aerosol types and the terminology defined inside the classification methods are presented in Table 1."

**Response to Reviewer #3:**

The authors would like to thank the reviewer for carefully reading the manuscript and for providing constructive remarks to improve this paper. The replies to the comments are given in the paragraphs below. The referee comments are highlighted in blue, while our responses in black fonts. The text modifications in the revised manuscript are highlighted in red.

"Section 2.1 (Source classification analysis): An altitude threshold of the trajectory is applied per domain in order for the algorithm to decide whether emitted particles could be carried away. The time the air mass remains below this altitude limit is crucial though. Is it meaningful to also include such an additional threshold in the algorithm?"

The time the air mass remains below the altitude limit is the actual output of the SCAN algorithm and is discussed in Section 2.1-SCAN. As it is now, it is up to the user of SCAN to assess the results of the classification. However, in the future we are planning on implementing additional thresholds for improving the algorithm.

"Section 2.1 (Source classification analysis): Is a single hot spot sufficient to mark a layer as smoke mixture? The authors could fine tune this limit (number of hot spots near the trajectory within a polluted/clean continental box" looking at the performance of SCAN in comparison with the other two algorithms"

The classification process of SCAN concerning the smoke aerosols is as follows: The latitude, longitude, height and time information of the trajectory are studied. If the latitude and longitude are found in a clean continental or polluted continental domain and if the height of the trajectory is lower than 3 km then the MODIS hot spots with confidence > 80% of this specific hour are studied. If a hot spot is found to have a distance < 8 km from the position of the trajectory at this hour, then this hour is marked as smoke. If more than 1 hour out of the 144 hours of the trajectory is found to have smoke contribution to the aerosol layer then the layer is characterized as smoke (or smoke mixture, depending on the presence of the other aerosols).

It is, also, possible that more than one hot spot to be present inside the 8 km distance from the trajectory at a specific hour. Finally, the pixel size of MODIS image is 250x250 m presenting the worth mentioning fire spots. Thus, with the addition of the confidence criteria we believe the fire identification of SCAN is accurate.

"Section 2.1 (Source classification analysis): The authors should add a few lines here to explain the link between the aerosol type and the 4 kinds of domains. While this is straightforward for marine and dust types, this is not the case for the continental polluted aerosols. The polluted continental domain on the map probably correspond to regions with increased urban activity but this has to be specified in the manuscript."

The first paragraph of this section was improved.

Initial: "SCAN is the automated aerosol layer classification process, optical property independent and developed in the IDL programming language in the frame of this study. For each identified aerosol layer a X-hours HYSPLIT backward trajectory (Draxler et al., 1998) is used to calculate the amount of time travelled above predefined aerosol source regions arriving over a lidar station at the specific date and height that the aerosol layer is observed. X is the number of hours of the backward trajectory which can be decided by the user at the beginning

of the process. SCAN assumes specific regions (Fig. 1, coloured squares, from now on mentioned as domains) in terms of aerosol sources (Penning de Vries et al., 2015)."

Changed: "SCAN is the automated aerosol layer classification process, optical property independent and developed in the IDL programming language in the frame of this study. For each identified aerosol layer a X-hours HYSPLIT backward trajectory (Draxler et al., 1998) is used to calculate the amount of time travelled above predefined aerosol source regions arriving over a lidar station at the specific date and height that the aerosol layer is observed. X is the number of hours of the backward trajectory which can be decided by the user at the beginning of the process. SCAN assumes specific regions (Fig. 1, coloured squares, from now on mentioned as domains) in terms of aerosol sources (Penning de Vries et al., 2015). The polluted continental domains represent the regions with increased anthropogenic activity, according to monthly means of tropospheric NO2 from GOME-2 (Georgloulias et al., 2019)."

"Section 2.1 (Neural network aerosol classification algorithm): Have the authors checked the performance of NATALI by applying different user configurations? Pieces of information such as the selected confidence level and minimum agreement threshold should be provided here."

Different configurations were selected for each case so that the geometrical characteristics were consistent with those retrieved manually.

"Section 2.3 (Case studies): A lot of numbers are provided in this part which makes it hard to follow. It would be more efficient if they were presented in tables."

A Table was added.

Initial: "The mean values of the intensive aerosol optical properties within the aerosol layer observed over Kuopio are: $lr_{355}$=65.58±11.02 sr, $lr_{532}$=72.51±17.61 sr, $A_{e355/532}$=1.23±0.62, $A_{b355/532}$=1.36±0.0.05, $A_{b532/1064}$=1.23±0.05 and $pldr_{532}$=2.1±0.1 %, indicating fine absorbing aerosols. The mean values of the intensive aerosol optical property within the lower aerosol layer observed over Potenza on the same day are: $lr_{355}$=35.97±1.09 sr, $lr_{532}$=24.55±4.15 sr, $A_{e355/532}$=1.14±0.44, $A_{b355/532}$=0.16±0.04, $A_{b532/1064}$=0.97±0.04, and $pldr_{532}$=13.5±0.4 %. Similarly, for the middle aerosol layer observed over Potenza: $lr_{355}$=31.05±2.11 sr, $lr_{532}$=22.50±1.69 sr, $A_{e355/532}$=0.86±0.25, $A_{b355/532}$=0.06±0.13, $A_{b532/1064}$=0.79±0.06 and $pldr_{532}$=15.4±1.5 %. These values of both lower and middle aerosol layers are indicative of coarse semi-depolarizing aerosols, probably, mixed dust particles. Finally, for the upper aerosol layer observed over Potenza: $lr_{355}$=38.77±4.81 sr, $lr_{532}$=24.44±3.39 sr, $A_{e355/532}$=0.56±0.37, $A_{b355/532}$=-0.58±0.25, $A_{b532/1064}$=0.72±0.07 and $pldr_{532}$=24.8±1.0 %, indicating coarse high-depolarizing aerosols, probably of dust origin."

Changed: "The mean values of the intensive aerosol optical properties within the aerosol layer observed over Kuopio and Potenza on 30 July 2015 are also presented in Table 3. In particular, for the case of Kuopio, the $lr_{355}$ values were found to be lower than those of $lr_{532}$, the Ångström exponent (both extinction- and backscatter-related) was higher than 1.2 and pldr532 had low values (<5 %), indicating fine absorbing aerosols. For the case of Potenza, the lr values were found to be low (<39 sr at 355 nm and <25 sr at 532 nm), while the Ångström exponents remained mainly below 1.0 for all the three aerosol layers observed. The difference between these three aerosol layers is the value of $pldr_{532}$ which was found to be ascended from 13.5±0.4 % (bottom layer) to 15.4±1.5 % (middle layer) and finally reached the value of 24.8±1.0 % (top

layer), indicating coarse semi-depolarizing aerosols at lower altitudes (<4.5 km) and high-depolarizing aerosols higher, probably of dust origin.

Table 3: Mean values of the intensive optical properties of aerosol layers observed on 30 July 2015 over Kuopio and Potenza.

| Site | Height [km] | $lr_{355}$ [sr] | $lr_{532}$ [sr] | $A_{e355/532}$ | $A_{b355/532}$ | $A_{b532/1064}$ | $pldr_{532}$ [%] |
|------|-------------|------------------|------------------|-----------------|-----------------|------------------|-------------------|
| Kuopio | 1.5-1.9 | 65.58±11.02 | 72.51±17.61 | 1.23±0.62 | 1.36±0.05 | 1.23±0.05 | 2.1±0.1 |
| Potenza (bottom) | 2.8-3.1 | 35.97±1.09 | 24.55±4.15 | 1.14±0.44 | 0.16±0.04 | 0.97±0.04 | 13.5±0.4 |
| Potenza (middle) | 3.4-3.9 | 31.05±2.11 | 22.50±1.69 | 0.86±0.25 | 0.06±0.13 | 0.79±0.06 | 15.4±1.5 |
| Potenza (top) | 4.5-5.4 | 38.77±4.81 | 24.44±3.39 | 0.56±0.37 | -0.58±0.25 | 0.72±0.07 | 24.8±1.0 |

"

"Line 111: Please specify also the accumulated probability that corresponds to a MD of 4.3 with 4 degree of freedom (independent variables). In addition, the authors explain why these thresholds are applied. Are they supported by previous studies?"

This method and the thresholds used are discussed at Papagiannopoulos et al., 2018. This reference was added at this point.

Initial: "In this study, we used four aerosol intensive properties: the backscatter-related Ångström exponent at 355 and 1064 nm, the aerosol lidar ratio at 532 nm, the color ratio of the lidar ratios and the aerosol particle linear depolarization ratio at 532 nm, whie the minimum accepted distance was set to 4.3. As soon as the distance from a specific aerosol class is below the threshold and the remaining distances are higher than the threshold, the observation is assigned to that aerosol class. If more than one distance is below the threshold, the normalized probability of each class needs to be over 50%."

Changed: "In this study, we used four aerosol intensive properties: the backscatter-related Ångström exponent at 355 and 1064 nm, the aerosol lidar ratio at 532 nm, the color ratio of the lidar ratios and the aerosol particle linear depolarization ratio at 532 nm, while, the minimum accepted distance was set to 4.3 (Papagiannopoulos et al., 2018). As soon as the distance from a specific aerosol class is below the threshold and the remaining distances are higher than the threshold, the observation is assigned to that aerosol class. If more than one distance is below the threshold, the normalized probability of each class needs to be over 50%."

"Line 168: This technique is also applied in NATALI. Is the layering automatically performed by SCAN or is the analysis based on layers obtained by NATALI? In the latter case, it has to be specified in the manuscript (e.g. "Source classification analysis" Section) that only the classification part of SCAN is automated, and not the layering part."

This was mentioned at the first paragraph of Section 2: "SCAN is the automated aerosol layer classification process, optical property independent and developed in the IDL programming language in the frame of this study."

The layering was performed by two different ways: 1) manually, 2) NATALI in order to evaluate the performance of NATALI, while different thresholds during the classification of NATALI were used so that the geometrical characteristics resulted by NATALI are in good agreement with those calculated manually. The automatically layering will be also included in SCAN in future study.

"Lines 224-251: Care has to be taken here because the Mahalanobis distance algorithm utilizes both two different probability metrics, the chi-squared probability (Mahalanobis Distance) and also the normalized probability. Which one are the authors referring to?

This paragraph was modified.

Initial: "In Fig. 5 we present the classification of the aerosol layers under study, by (a) NATALI, (b) MD and (c) SCAN. The "aerosol type" results are given, concerning the classification by NATALI. Concerning the MD classification, the possibilities of each aerosol type assumed by MD are, also, given. Finally, concerning the classification of the aerosols by SCAN, the time (in hours) within which the air mass circulated over specified domains is also provided. It should be noted here that different colours refer to different aerosol types or aerosol mixtures.".

Changed: "In Fig. 5 we present the classification of the aerosol layers under study, by (a) NATALI, (b) MD and (c) SCAN. The "aerosol type" results are given, concerning the classification by NATALI. Concerning the MD classification, the normalized probabilities of each aerosol type assumed by MD are, also, given. Finally, concerning the classification of the aerosols by SCAN, the time (in hours) within which the air mass circulated over specified domains is also provided. It should be noted here that different colours refer to different aerosol types or aerosol mixtures."

"Line 28: "...while MD has the percentage of..." Do the authors mean highest percentage?"

Yes, the word "higher" was added.

Initial: "while MD has the percentage of unclassified layers (50%) and the lower percentage of cases classified as aerosol mixtures (5%)."

Changed: "while MD has the higher percentage of unclassified layers (50%) and the lower percentage of cases classified as aerosol mixtures (5%)."

"Line 55: Please correct the typo "neither objective nor automated""

This was corrected

Initial: "However, this case-by-case aerosol layer identification is not objective automated."

Changed: "However, this case-by-case aerosol layer identification is neither objective nor automated."

"Line 117: Please replace "3+2" with "3b_λα+2a_λα""

This was replaced.

Initial: "Their study used a 3b+2e lidar configuration"

Changed: "Their study used a 3b$_{λα}$+2e$_{λα}$ lidar configuration"

"Figure 3: I suggest that the authors use different colors for the layer base and top. Otherwise the layer boundaries can be confusing for the Extinction and Backscatter profiles."

Figure 3 was improved.

[Figure]

"Line 423: "dust and dust aerosols mixtures" Is this a typo?"

This sentence was rephrased.

Initial: "Concerning the MD's "unknown" and NATALI's "mixtures" categories, MD algorithm revealed an increased contribution of dust and dust aerosols mixtures (approximately 50%) inside the studied aerosol layers."

Changed: "Concerning the MD's "unknown" and NATALI's "mixtures" categories, MD algorithm revealed an increased contribution of dust aerosols (approximately 50%) inside the studied aerosol layers."

"Figure 5: Please check the percentages of pie charts of Figure 5 as the do not sum up to exactly 100%."

This figure was improved.

[Figure]

"Figure 6: Please check the percentages of pie charts of Figure 6 as the do not sum up to exactly 100%."

This figure was improved"